# An updated suite of viral vectors for in vivo calcium imaging using intracerebral and retro-orbital injections in male mice

Sverre Grødem[1,2], Ingeborg Nymoen [1,2], Guro Helén Vatne[1], Frederik Sebastian Rogge[1], Valgerður Björnsdóttir[1], Kristian Kinden Lensjø [1,3] ✉ & Marianne Fyhn[1,3]

Genetically encoded $Ca^{2+}$ indicators (GECIs) are widely used to measure neural activity. Here, we explore the use of systemically administered PHP.eB AAVs for brain-wide expression of GECIs and compare the expression properties to intracerebrally injected AAVs in male mice. We show that systemic administration is a promising strategy for imaging neural activity. Next, we establish the use of EE-RR- (soma) and RPL10a (Ribo) soma-targeting peptides with the latest jGCaMP and show that EE-RR-tagged jGCaMP8 gives rise to strong expression but limited soma-targeting. In contrast, Ribo-tagged jGCaMP8 lacks neuropil signal, but the expression rate is reduced. To combat this, we modified the linker region of the Ribo-tag (RiboL1-). RiboL1-jGCaMP8 expresses faster than Ribo-jGCaMP8 but remains too dim for reliable use with systemic virus administration. However, intracerebral injections of the RiboL1-tagged jGCaMP8 constructs provide strong $Ca^{2+}$ signals devoid of neuropil contamination, with remarkable labeling density.

The use of microscopy to measure the activity of neurons is widely applied in modern neuroscience. With the development of genetically encoded $Ca^{2+}$ indicators (GECIs) there have been rapid advances in response kinetics, sensitivity, and brightness of $Ca^{2+}$ sensors (e.g.,[1–3]), of which the GCaMP sensors are the most prominent. These engineered proteins contain a $Ca^{2+}$-binding motif and a circularly permuted green fluorescent protein that brightens when $Ca^{2+}$ is present. Using GECIs for activity measurements allows for cell-type targeted recordings, repeated measurements of the same cells for up to several months, and recordings from large populations of neurons. Ideally, GECIs should be uniformly expressed across the neuronal population with sufficient activity-dependent sensitivity and fluorescence range for in vivo applications. Overexpression in a subset of cells can lead to intracellular aggregation of GCaMP which can affect cellular function, and ultimately lead to cell death.

Various methods to introduce genetic constructs into cells have been explored. An often-preferred method for introducing GECIs into neurons is using transgenic animal models that are genetically modified to express the desired construct (e.g., [4–6]). These models allow for strong, evenly distributed, and sustained expression throughout the animal's life and can be targeted to specific cell types. However, transgenic GCaMP mice require intricate and costly breeding schemes, both financially and for animal welfare. Furthermore, they depend on driver lines that prevent the simultaneous use of other transgenics[3], thus limiting experimental flexibility. Moreover, since the GECIs are expressed throughout development, this may account for the frequent ictal activity that has been reported for several such mouse strains[7], questioning their reliability for measurement of neuronal activity. Finally, relying on transgenic GECI-expressing lines limits the opportunity to adopt new and improved GECIs as they become available. New GECIs with faster kinetics and higher sensitivity or different excitation/emission wavelengths are frequently published, while developing transgenic lines takes considerably longer, and adopting new lines comes with substantial costs.

An alternative strategy is to deliver GECIs to neurons using a viral vector, which, in contrast to transgenic lines, allows for complete

[1]Center for Integrative Neuroplasticity, Department of Bioscience, University of Oslo, Oslo, Norway. [2]These authors contributed equally: Sverre Grødem, Ingeborg Nymoen. [3]These authors jointly supervised this work: Kristian Kinden Lensjø, Marianne Fyhn. ✉e-mail: kristian.lensjo@ibv.uio.no

flexibility in the adoption of alternative GECI constructs. In the brain, viral vectors encoding GECIs can be delivered by way of an intracerebral (IC) injection directly into the tissue of interest (e.g.,[8]) or cerebral ventricles[9]. However, IC injections of viral vectors tend to lead to highly variable expression depending on the concentration of virus particles and are often associated with cell damage or death[10]. Furthermore, using adeno-associated virus (AAV) serotype 9 which crosses the blood-brain-barrier[11] in neonatal mice, GECIs may be administered systemically through an intravenous injection into the tail -or temporal vein[12], or the transverse sinus[13]. However, these administration techniques are technically challenging, and come with a high risk of overexpression of the GECI because of the young age at the time of injection, and extended period from injections to experiments which may lead to cell damage or ictal events.

In contrast to AAV9, the recently developed AAV serotype PHP.eB crosses the blood-brain barrier in adult animals and efficiently transduces neurons across the brain[14], suggesting that genes can be delivered via intravenous injections[15]. Importantly, such injections can be performed at any stage of development and thus prevent accumulation of GECIs and disturbances to Ca$^{2+}$ homeostasis during sensitive parts of development. Moreover, this would enable brain-wide expression of the GECI in combination with transgenic models for e.g. cell-type specific activity perturbations. In contrast to the tail vein and other intravenous injection procedures, injections into the retro-orbital (RO) sinus can be performed with only limited training. RO injections are quick, non-invasive, and impose little stress on the animals compared to other methods[16].

With the recent development of new GECIs and methods to restrict expression to parts of the cell, in combination with alternative delivery methods, there is need for a systematic assessment of these approaches. Here, we present a screening of multiple GECI constructs in mice comparing the RO injection method for systemic viral delivery with IC injections in primary visual cortex (V1), and assess functionality of the GECIs using wide-field and two-photon laser-scanning microscopy. We screen both widely used GECIs, such as GCaMP6f, and recently developed GECIs with improved sensitivity and kinetics[3] (jGCaMP8s and jGCaMP8m). We show that several modern GECIs are highly suitable for systemic administration and give rise to uniform and stable expression for many weeks, and can readily be combined with other transgenic models for e.g., cell-specific expression of optogenetic or chemogenetic receptors. Because of the high brightness and sensitivity of jGCaMP8s, we also apply EE-RR- and ribosome-targeting peptides, where GECI localization is restricted to the soma, in order to limit neuropil signal[17,18]. Ribosome-tethered (RL10) jGCaMP8 provided highly selective expression in the cell soma and showed remarkable density of cell labeling for two-photon imaging, but the expression rate was substantially reduced compared to EE-RR soma targeting and non-soma-targeted GECIs. To circumvent this issue, we screened three different linker peptides in an effort to improve expression rate. One of these linker peptides, a long and flexible GS repeat linker, provided strong expression just one week after intracerebral injection. The ribosome-tethered construct with a modified linker, RiboL1-jGCaMP8, is both rapidly expressed and is all but completely excluded from neuropil.

## Results

In order to compare the performance of GECI variants with different administration methods, we initially screened the performance of 10 existing GECI variants administered by RO or IC injections in high titer PHP.eB serotype AAVs, all expressed under a synapsin promoter. Pairs of mice were randomly assigned a GECI-expressing AAV and evaluated every two weeks for 2.5 months using wide-field and two-photon imaging through a cranial window. Expression of the GECIs were confirmed by post-mortem immunohistological inspection (Figs. 1a–c, S1, S2). In vivo imaging was performed in awake, head-fixed mice

running freely on a running wheel. We collected imaging data during periods of spontaneous activity (in darkness) or during presentation of visual stimuli (drifting sinusoidal gratings) for all GECIs tested. We mainly focused the two-photon imaging to layer 2/3 neurons at a depth of approximately 150–200 μm below the cortical surface, but images were also captured from deeper layers (3–400 μm, Fig. 1d) to assess expression levels across layers. The example images and analysis shown in the following section are based on data collected during spontaneous activity, unless described otherwise.

Of the GECIs included in our initial screening (Table 1), which are all variations of GCaMP, most have been widely used with IC injections in previous work. We confirmed detectable expression at reasonable laser power (40–50 mW output power at the front aperture of the objective at 920 nm) when administering the GCaMPs with IC injections with the PHP.eB serotype. However, following systemic virus administration, the majority of these GECIs were not sufficiently bright for in vivo Ca$^{2+}$ imaging (Fig. 2a, Table 1). The most widely used GECI, GCaMP6f, was undetectable using reasonable laser power when expressed from an RO injection. At 6 weeks post injection, GCaMP6f was detectable, but only at very high laser power (>140 mW output at the objective) which would not be sustainable for functional experiments (Fig. 2a). Moreover, jGCaMP8f, which is reported to be brighter than previous "fast" iterations, was also not sufficiently bright for in vivo imaging when administered systemically. In an attempt to improve the brightness, we both doubled and tripled injection volumes of RO administered GCaMP6f and jGCaMP8f, but the resulting expression was still too dim to image at reasonable laser power (in vivo data for triple injection volumes were of similar brightness to the image of GCaMP6f shown in Fig. 2a, histology shown in Fig. 2b, c). Of the more recently developed GECIs; jGCaMP7s, jGCaMP8s, and jGCaMP8m were all sufficiently bright for use with RO-injected viruses (Figs. 1c and 2a, Supplementary Video 1), i.e., single-cell Ca$^{2+}$ transients were detectable at reasonable laser power (40–50 mW). Of these, jGCaMP7s displayed the lowest neuropil signal but also the slowest response kinetics, the latter in line with earlier reports[2]. Encouragingly, none of the GECIs we tested by RO injection showed signs of ictal activity (measured by wide-field fluorescence imaging, example data shown in Fig. S1), contrasting previous reports in transgenic GECI-expressing mice[7].

Importantly, post-mortem histological analysis indicated that brightness of the GECI was the determining factor for whether a GECI was detectable in vivo, as the expression of GCaMP when labeled with a GFP antibody was comparable between GECIs with very different in vivo performance (Figs. 2b and S3a–c). Overall, the histology and in vivo imaging matched previous reports on PHP.eB infection[19], both with respect to the expression pattern across the cortex and that the highest density was found in cortical layer 5 (Figs. 1d, 3a, S2 and S3). This was true for the visual cortex, somatosensory, retrosplenial, and motor cortex (Fig. S2). In the hippocampus, we observed only sparse labeling, apart from very dense labeling in area CA2 (Figs. S2 and S3). We also confirmed expression in the spinal cord (Fig. S2, lower panel).

While sufficient brightness for imaging is a requirement for any GECI to be viable, there are many factors to consider when selecting the optimal sensor for a given experiment. The newer iterations of GCaMP feature improved kinetics and higher sensitivity relative to previous versions. However, higher sensitivity may also affect the signals detected from neuronal processes such as dendrites, which can both disturb the quantification of Ca$^{2+}$ transients from the cell soma and reduce the number of cells one can record from by introducing noise. Recent efforts have attempted to alleviate these effects by restricting the expression of the GECI to the cell soma[17,18], but it is unclear how the soma-specificity and efficiency are affected by brain-wide expression of the brighter iterations of jGCaMP. We therefore applied two main soma-targeting strategies using jGCaMP8 and compared their performance with IC and RO-injected viruses.

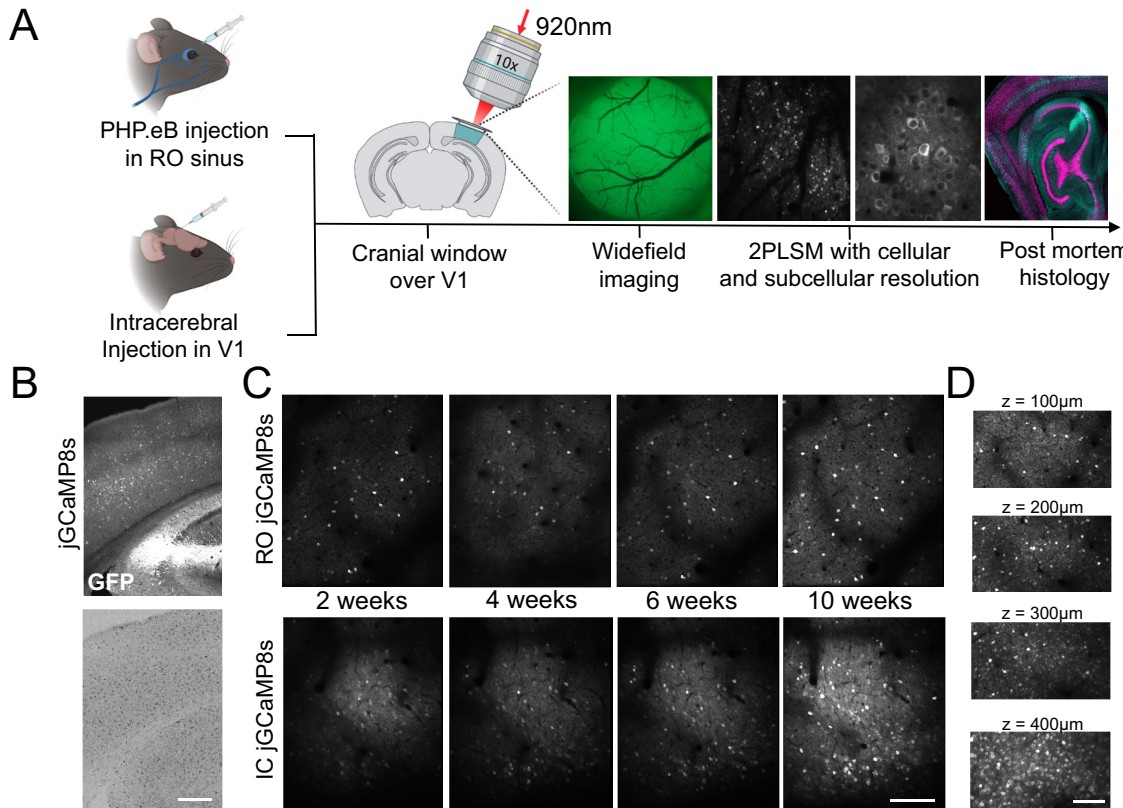

**Fig. 1 | Systemic and local administration of GECIs using PHP.eB AAVs.**
**A** Experimental overview indicating the two injection methods, and the approaches used to monitor the expression. Illustration created with Biorender.com. **B** Post-mortem histological verification of jGCaMP8s expression 10 weeks after RO injection and microglia activation verified by Iba1-positive labeling. Scale bar indicates 200 μm. **C** Example images from in vivo two-photon microscopy of jGCaMP8s expressed by RO or IC virus injections 2, 4, 6, and 10 weeks after injection. Scale bar indicates 200 μm. **D** GECI expression at different depths in cortex after RO injection of jGCaMP8s. Scale bar indicates 150 μm.

We first constructed EE-RR tagged versions of the jGCaMP8 variants that showed functional expression levels after systemic virus injection (jGCaMP8m and s). 2–4 weeks post RO injection, the EE-RR-jGCaMP8 was comparably bright to the unaltered jGCaMP8 but had limited effects on the localization to neuropil (Figs. 3a, S3,

**Table 1 | Overview of initial GECI screening with virus titer and Addgene reference indicated**

| GECI | Titer (VG/ml) | Detectable at 50 mW (RO injection) | Addgene plasmid # |
|---|---|---|---|
| GCaMP6f | 2.44E + 13 | No | 100837 |
| EE-RR- GCaMP6f | 2.04E + 13 | No | 158756 |
| jGCaMP7f | 2.04E + 13 | No | 104488 |
| EE-RR- GCaMP7f | 1.53E + 13 | No | 158760 |
| jGCaMP7s | 1.11E + 13 | Yes | 104487 |
| jGCaMP8f | 1.04E + 13 | No | 162376 |
| jGCaMP8m | 9.43E + 12 | Yes | 162375 |
| jGCaMP8s | 1.49E + 13 | Yes | 162374 |
| jREX-GECO1 | 9.36E + 12 | Yes | 169259[a] |
| Ribo-GCaMP6m | 1.86E + 13 | No | 158777 |
| CAG-mNeonGreen | 1.09E + 13 | Yes | 99134 |

[a]jREX-GECO1 expressed from a hSyn promoter was made and used in this manuscript.
All constructs were tested and confirmed viable for imaging using IC injections (with the exception of Ribo-GCaMP6m, where we did not observe in vivo expression). "Detectable at 50 mW (RO injections)" is defined as whether a single imaging plane using a 16× Nikon objective and 50 mW laser power at 920 nm gives clear single-cell fluorescence transients 6 weeks after injection. The fluorescent protein mNeongreen expressed under a CAG promoter was used as a positive expression control for the RO injection method.

Supplementary Video S2 and 3). Similar to the RO-injected animals, we observed strong signals from both EE-RR and regular jGCaMP8 after IC injections with comparable neuropil signals (Fig. 3b, Supplementary Video S4).

In comparison to EE-RR soma targeting and other non-soma-targeted GECIs, ribosome-tethered (RPL10a) GCaMP expression has been reported to drastically reduce the brightness of the attached GCaMP[17], demanding high laser intensity for in vivo imaging. In contrast to EE-RR, Ribo-GCaMP localization is strictly confined to the soma. We therefore constructed Ribo-jGCaMP8m and Ribo-jGCaMP8s and first tested their suitability for systemic virus injections. In line with the previously reported reduction of brightness of GCaMP6m by the Ribo-tag, Ribo-GCaMP8s only displayed dim signal confined to a small space in the soma 2 and 4 weeks after injection. After 6 weeks the signal had improved somewhat, to a level where single cells could be observed, but was still dim relative to the other functional constructs (Fig. 3). The expression was strictly confined to the soma, with little to no visible neuropil signal (Figs. 3a, right panel, S3b), however, we also observed indications of aggregated GCaMP in RO injected animals, possibly from projecting axon terminals or small ribosome clusters in dendrites. When delivered by an IC injection, Ribo-jGCaMP8 was relatively dim 2 and 4 weeks after injection and only showed small aggregated fluorescent spots with static signals (Fig. 3c, left image), similar to what we observed after RO injections. However, 6 weeks after IC injection both Ribo-jGCaMP8m and Ribo-jGCaMP8s showed bright and dynamic signals (Figs. 3b, S3b, Supplementary Video S5 and S6). The expression remained remarkably stable over time, up to the last sampling point 10 weeks post injection.

Notably, the wide-field signals from mice injected with Ribo-jGCaMPs were very weak throughout the experiment, likely as a result

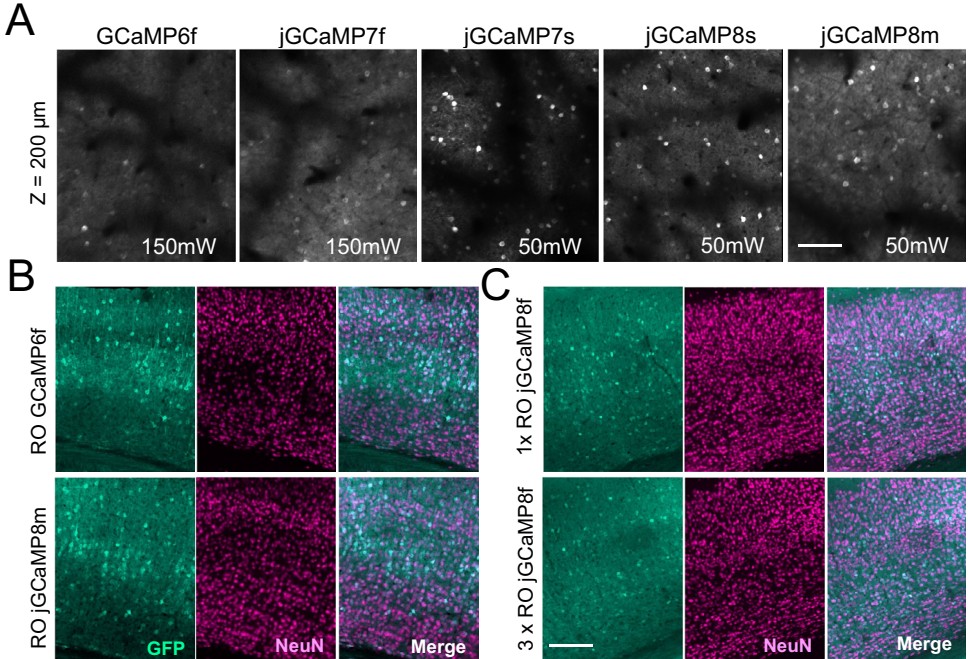

**Fig. 2 | In vivo GECI screening after retro-orbital (RO) virus injections.**
**A** Representative example images acquired by two-photon microscopy from five different GECIs acquired six weeks after RO virus injections. Note that GCaMP6f and jGCaMP7f were not detectable at reasonable illumination intensity (40–50 mW at 920 nm). The images shown from GCamp6f and jGCaMP7f were acquired using 150 mW laser power for testing purposes only. The recently developed jGCaMP7s, 8 s, and 8 m were sufficiently bright using 50 mW. Scale bar indicates 250 µm. All images are average intensity projections from 2000 frames with identical adjustments to brightness and contrast. **B** Representative histology images of sections from the primary visual cortex six weeks after RO injection. **A**, **B** the results were reproduced in $n \geq 3$ mice. **C** Similar to **B**, showing a comparison of expression levels of jGCaMP8f between RO injections of 100 or 300 µL volume. Scale bar for **B**, **C** indicates 250 µm. The results were reproduced in 2 mice.

of the lacking neuropil signals (example image from Ribo-jGCaMP8s shown in Fig. S1a). This indicates that wide-field $Ca^{2+}$ imaging using these sensors would require far more sensitive imaging equipment compared to non-targeted or EE-RR-targeted jGCaMP8, but the recorded signal would reflect activity in the cell soma and not e.g. projecting axons.

We also tested former iterations of GCaMP (6m, 6f, 7f) in combination with EE-RR or ribosome-tethering. As expected from the performance of non-soma-targeted versions, these were not bright enough to use with RO injections.

The slow expression and aggregation of Ribo-jGCaMP8 that we observed may be a limiting factor to some experiments, for example by preventing imaging experiments in young animals, requiring removal of bone growth in suboptimal cranial window implants, or having to perform the virus injections and window implant in separate surgeries. In an attempt to improve the rate of expression, we replaced the linker region of Ribo-GCaMP8 with three different linker sequences; one longer and more flexible sequence, and two variations of rigid, helical linkers. The rigid helical linkers failed to rescue Ribo-GCaMP8 expression. However, the more flexible and longer linker, which is identical in amino acid sequence to the one used in EE-RR-GCaMP, greatly increased the rate of protein expression. This construct, which we term RiboL1-jGCaMP8s, displayed strong expression 1–2 weeks after intracerebral virus injection (Fig. 3b, c, Supplementary Video S7). In contrast to the original Ribo-jGCaMP8 construct, we could detect a substantial number of single cells just two weeks after the injection, with no indication of the aggregated, static puncta that we observed at the same time-point using Ribo-jGCaMP (Fig. 3c). The expression remained stable across many weeks. When tested with systemic injections, RiboL1-jGCaMP8 (m and s versions) was relatively dim, similar to the original Ribo-jGCaMP8 construct (example shown Fig. 4). Because of the promising expression patterns of RiboL1 compared to

the former ribosome-tethered versions when used with IC injection, we proceeded with this construct for further testing and analysis.

To verify that the functional properties of the neurons expressing EE-RR-and RiboL1 versions of jGCaMP8 were not compromised, we presented the mice with visual stimuli and measured the responses of single neurons in V1. With both GECIs we found strong visual responses and stable orientation tuning (examples shown in Fig. S5). Quantifications of spontaneous and stimulus-evoked $Ca^{2+}$ events are shown in Fig. S5.

To directly compare the performance of the most promising GECIs using the two injection approaches, we first monitored expression stability over time, from 2 to 10 weeks after virus injection (Fig. 4). Our data indicated that RO injection gave rise to stable expression levels when compared to IC injection (Fig. 4), with no indication of microglia activation or intracellular aggregation (Fig. S6a). Notably, the RiboL1-jGCaMP8 construct gave rise to very stable expression levels when injected by an IC injection (Fig. 4, lower left panel) with no indication of intracellular aggregation. Indeed, in a separate experiment, we could record from the same population of neurons for more than 5 months (Fig. S6b).

Next, we compared the extracted $Ca^{2+}$ traces from the cell soma and neuropil for the most promising GECIs using both injection techniques. The data used for the analysis were acquired 4–6 weeks after virus injections, and we narrowed our analysis to include jGCaMP8s, EE-RR-jGCaMP8s, and RiboL1-jGCaMP8s. We also included mice with IC injections of GCaMP6f for comparison, and for a within-subject visual comparison with RiboL1-jGCaMP8s, as these two GECIs were injected in opposite ends of the very same cranial windows in three mice (Fig. S4d).

We first calculated correlation coefficients between the soma and neuropil signal of the different GECIs. As described above, the high sensitivity and brightness of jGCaMP8 may lead to neuropil signals that

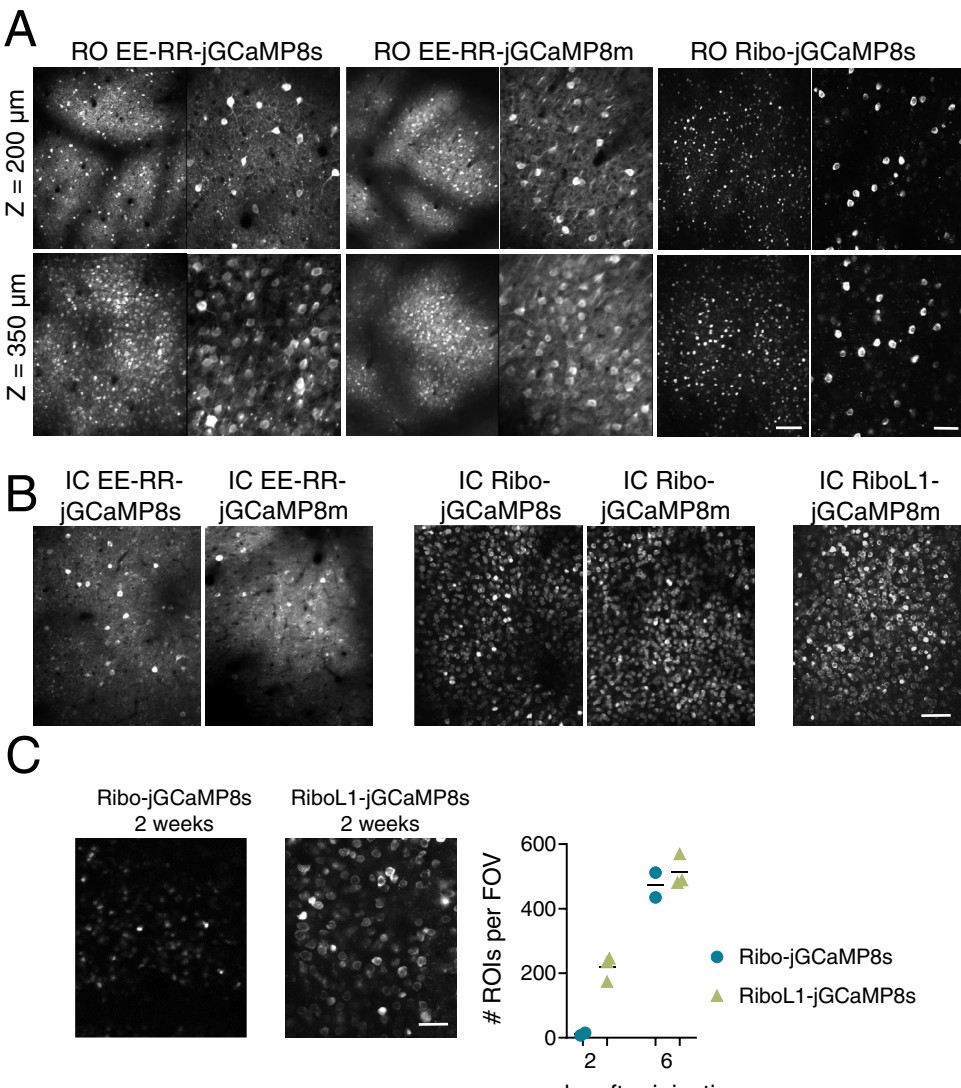

**Fig. 3 | Soma-targeted jGCaMP8 expressed by RO and IC injections in primary visual cortex. A** Representative example images from RO injected mice expressing EE-RR-jGCaMP8s and m, and Ribo-jcaMP8s, at two different depths in cortex. Scale bars on lower right aligned images indicate 150 and 50 µm, respectively. All images shown are average intensity projections from 2000 frames with identical adjustments to brightness and contrast. Data from EE-RR constructs was acquired 4 weeks after virus injections, and 6 weeks for the ribosome-tethered construct. **B** Similar to **A**, showing expression after IC injections. Note the high number of identifiable single neurons with ribosome-tethering, compared to jGCaMP and EE-RR-jGCaMP8. Scale bar indicates 100 µm. **C** Development of expression of Ribo-jGCaMP8s and RiboL1-jGCaMP8s after IC injections. Two weeks after virus injections, Ribo-jGCaMP8s was only visible as static spots, while with RIboL1-jGCaMP8s individual neurons were identified. Six weeks after injection the two constructs were expressed similarly, as shown in **B**. Right panel shows the number of identified neurons per field of view 2 and 6 weeks after virus injections, identified from average intensity projections of 250 imaging frames. Field of view size was 420 × 350 µm, and $n = 2$ and 3 mice for Ribo-jGCaMP and RIboL1-jGCaMP, respectively. Scale bar indicates 30 µm. **A**–**C** The results were reproduced in $n \geq 3$ mice.

skew the ΔF/F calculation ratio which is often used for measuring neuronal activity. We found that the soma and neuropil signals were highly correlated for jGCaMP8s and EE-RR-jGCaMP8s, independent of injection technique (Fig. 5a). In contrast, RiboL1-jGCaMP8s correlations to neuropil were low. We next measured the baseline brightness (Fig. 5b). In line with our earlier observations, the expression of jGCaMP8 gave rise to the brightest baseline signals (median values across a recording), and fields-of-view (FOVs) with an IC injection were brighter than from RO injections. Moreover, RiboL1 baseline fluorescence was low, in particular when expressed from RO injections. However, when we compared the ratio between soma and neuropil signal brightness at baseline, RiboL1-jGCaMP8s showed the highest ratio using both RO and IC injections (Fig. 5c). Notably, RO injections of jGCaMP8s and EE-RR-jGCaMP8s gave rise to higher ratios compared to IC injections of the same constructs. Together, these results indicate

that IC-injected RiboL1-jGCaMP8s may provide better estimates of fluctuations of Ca²⁺ dynamics in the cell soma compared to the other constructs tested.

We next investigated the expression of GCaMP within the individual FOVs for each GECI, by comparing images made from average intensity projections. Surprisingly, the number of cells identified was not very different between RO and IC injected jGCaMP8s and EE-RR-jGCaMP8s, despite the difference in concentration of virus particles in the tissue that arises from the two techniques. However, this difference was substantial for RiboL1-jGCaMP8s between the two methods. The IC injections of RiboL1-jGCaMP8s gave by far the highest number of identified cells, which is in line with a previous report on ribosome-tethering and our earlier observations (Figs. 3b, 5d, e, S3).

Virus delivery by IC injections distributes the virus unevenly in the tissue, with a higher concentration of viral particles closer to the

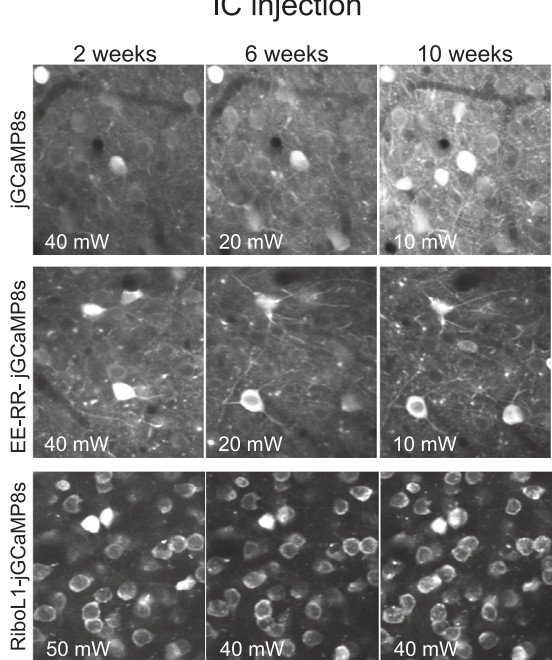
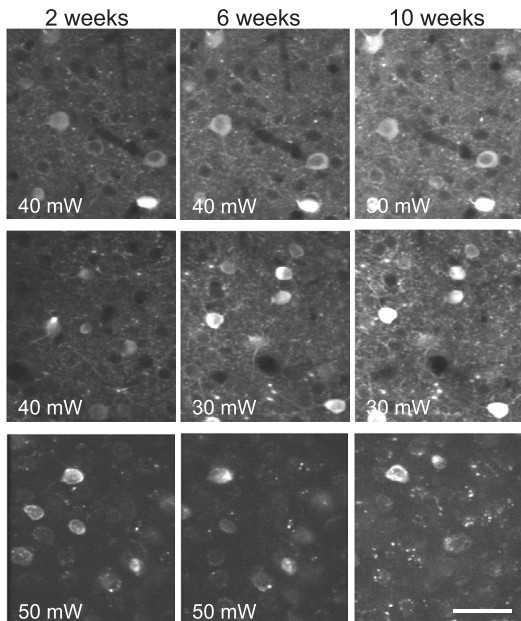

**Fig. 4 | Expression over time of jGCaMP8 and soma-targeted jGCaMP8 using RO and IC virus injections.** Note the dramatic reduction in illumination power (noted in the bottom left corner) used for jGCaMP8s and the EE-RR versions when expressed by an IC injection. RO injected jGCaMP8s and EE-RR-jGCaMP8s gave rise to very stable expression levels. For RiboL1-jGCaMP8s RO injections the signal was relatively weak, but when using the same construct with IC injections a large population of neurons could be detected and the signal intensity remained stable throughout the experimental period. All images shown are average intensity projections from 250 frames with identical adjustments to brightness and contrast, and the results were reproduced in ≥3 mice. Scale bar indicates 30 μm.

injection site. This may lead to differences in signal intensity across the imaging area, and again affect the quality of the data collection, an effect that may be alleviated by systemic administration of the viral vector. To test this, we compared the signal intensity across the FOV (x direction) at three evenly distributed sampling sites per FOV (Fig. 5e). We found that RO injections of jGCaMP8s indeed showed even signal intensity across the FOV, albeit with some variation and higher intensity around the central area, possibly reflecting uneven illumination that leads to increased neuropil signals. In contrast, the signal intensity after IC injections of jGCaMP8s was more than 40% higher in the center of the FOV compared to the edges. The expression following IC injections of RiboL1-jGCaMP showed by far the most even signal intensity, likely reflecting the almost complete lack of neuropil signals (Fig. 5f).

**Applications of systemic GECI injections**

One of the challenges with using transgenic mice to express a GECI is that it limits the use of cell-type targeted transgenics, e.g. manipulations of the activity of a specific cell population, while expressing the GECI brain-wide. To test the suitability of RO injections for this purpose, we used PV-Cre mice[19] that express Cre under the parvalbumin (PV) promoter and performed an RO injection of EE-RR-jGCaMP8s combined with intracerebral injections of an AAV5 vector expressing a FLEX-hM4D receptor (DREADD). Indeed, this produced co-expression of both constructs in putative PV neurons and with a uniform expression of jGCaMP8 in surrounding neurons (Fig. S7a). This was also verified by post-mortem histology (Fig. S7a).

Next, we tested the red-shifted GECI jREX-GECO1 (Fig. S7b), a long stokes-shift version of the red GECI RGECO that is optimized for two-color imaging with a single laser source. This enables imaging with the same 920 nm laser as for green GECIs meaning that jREX-GECO1 could be used for imaging axonal activity in combination with a soma-targeted green-shifted Ca$^{2+}$ indicator (e.g., [20,21]). To this end, we constructed axon-targeted jREX-GECO1 (hSyn-Axon-jREX-GECO1) and

performed virus injections into the dorsal lateral geniculate nucleus, combined with IC injections of RiboL1-jGCaMP8s in V1. We then imaged the activity in axonal boutons and cell somas in V1 two weeks after viral injections, and found strong signals from both GECIs (Fig. 6a, f shows examples from in vivo imaging and histology, respectively). We detected axonal boutons with both highly correlated and non-correlated changes in fluorescence with the soma signal (Fig. 6b–e). Axon-jGCaMP8s m and f were also prepared, but were not tested in this paper.

## Discussion

Engineered AAV serotypes with high affinity for the central nervous system that can be delivered intravenously represent a minimally invasive and low-cost method for introducing genetic payloads into the brain. Yet, these new serotypes, notably PHP.eB, have not been much used to deliver GECIs, despite obvious advantages in terms of animal welfare, cost, productivity, and experimental flexibility. Previous work shows that wide-field imaging with systemically administered GCaMP6f is feasible using other promoters than synapsin[15,22]. In our own preliminary experiments, we experienced that widely used GECIs (such as GCamp6f) was not bright enough to be compatible with systemic administration for use with two-photon imaging. A reduction in brightness in a systemic administration is not unexpected, as the multiplicity of infection will be lower relative to an intracerebral injection, i.e., each cell is infected by fewer viral particles. Here, we screened 16 GECIs and one fluorescent probe and show that newer iterations of jGCaMPs, particularly jGCaMP7s, jGCaMP8s, and jGCaMP8m, are indeed sufficiently bright for two-photon in vivo Ca$^{2+}$ imaging when administered systemically in PHP.eB AAVs. However, we also observed strong neuropil signals from these sensors which may influence the detection accuracy of individual neuron activity, regardless of the administration route. By fusing the latest jGCaMP variants with soma-targeting peptides we overcame this issue. We show that

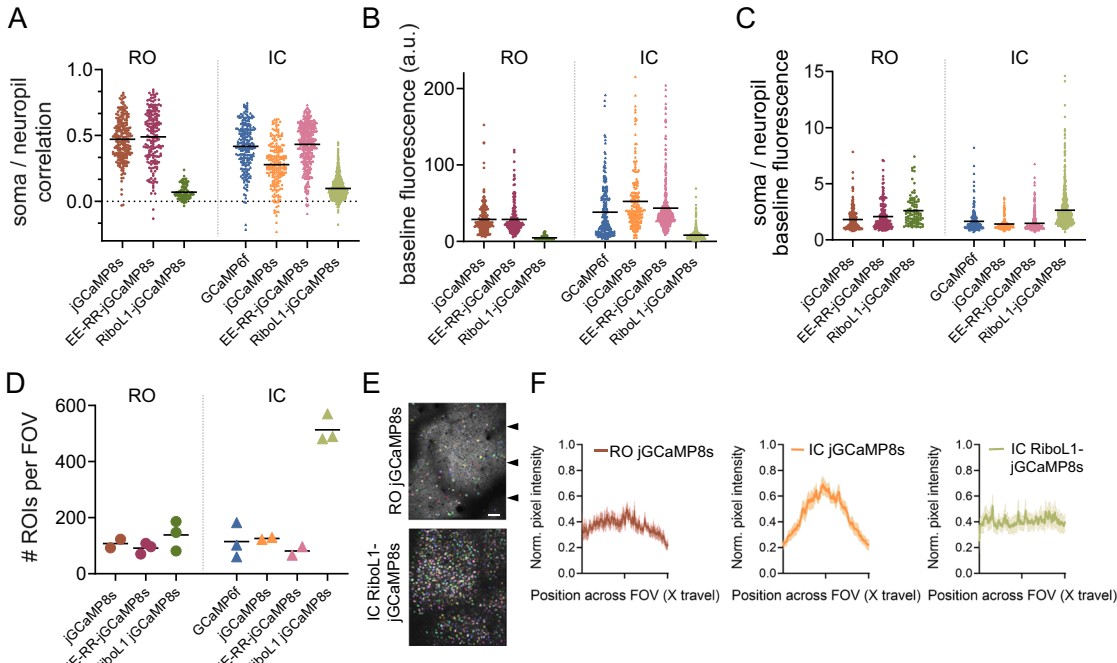

**Fig. 5 | Baseline brightness, signal-to-noise, and cell density quantification.**
**A** Pearson correlation coefficients of somatic to neuropil signals of RO or IC administered GCaMP. **B** Baseline (median) fluorescence intensity. **C** Soma over neuropil baseline (median) fluorescence intensity. **A**–**C** $n = 4$ mice per GECI for RO injections, 3 mice per GECI for IC injections; all data points indicate individual cells, lines indicate population means. **D** Number of ROIs detected per field of view in Cellpose. Each data point indicates one FOV from one mouse, line indicates

population mean. Scale bar indicates 50 μm. **E** Examples of Cellpose segmentation mask (ROIs) for RO jGCaMP8s and IC RiboL1-jGCaMP8s. Arrows indicate the y positions of measurements used for **F**. **F** Normalized intensity profiles across the field of view. Intensity was measured at three $y$ positions and normalized to the highest pixel value. $n = 5$, 4, and 4 mice. Lines indicate population mean, shaded area represents the SEM. Source data are available as a Source Data file.

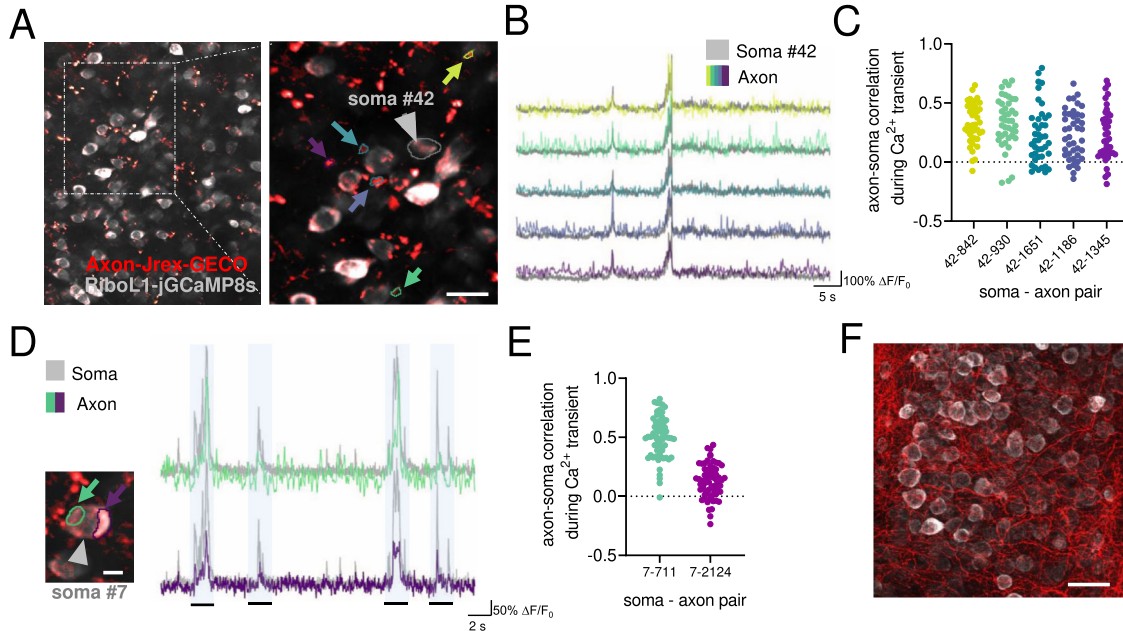

**Fig. 6 | Simultaneous in vivo imaging of axonal and somatic activity. A** An example field-of-view in primary visual cortex with Axon-jREX-GECO1 in red and RiboL1-jGCaMP8s in gray. Colored arrows indicate labeled regions of interest from the red channel. The cell soma ROI from the green channel is indicated by a gray arrow. Scale bar indicates 15 μm. **B** Red channel Ca²⁺ traces (colored) from each bouton overlaid on the soma signal (gray). Despite clear spatial separation, the indicated regions of interest show similar activity patterns (middle and right panel). **C** Cross-correlation of soma and axon signal during periods of Ca²⁺ transients in the

soma. Each dot represents the correlation between soma # 42 (as shown in **B**) and a specific axon bouton (colors indicate different boutons shown in **B**), for each somatic Ca²⁺ transient during a recording. **D**, **E** Two regions of interest in close proximity to a cell soma, indicated by colored arrows, and corresponding Ca²⁺ traces overlaid on the soma signal at different time points. The shaded regions represent time bins where cross-correlations were calculated. Scale bar indicates 5 μm. **E**, **F** Histological verification of expression. Source data used for C and E are available as a Source Data file. Scale bar indicates 25 μm.

RiboL1-jGCaMP8 outperforms existing GECI constructs using intracerebral virus injections.

An intravenous AAV injection in the retro-orbital sinus can be performed within minutes and requires very little training. The procedure is substantially less invasive than stereotaxic/intracerebral injections. The intravenous injection provides largely uniform expression across the mouse brain, which is stable over extended time periods. This stands in contrast to intracerebral virus injections that may result in excessive expression leading to unhealthy cells and even cell death. Moreover, we observed that the variability in expression that may arise from differences in viral titers is less prominent with RO-administrated AAVs compared to IC administration. This is in line with our observations from the triple RO injected animals, where more virus delivered across days did not increase GECI expression. We note that in our hands, stable expression was resulting both from the administration route and the PHP.eB serotype.

Despite the many advantages of intravenously injected virus for GECI delivery, it has one considerable drawback; intravenous administration requires a large dose of virus per animal. If all viruses are purchased from commercial vendors, this could be prohibitively expensive. On the other hand, if viruses are produced in-house or by a local virus core, scaling up production to suitable levels is relatively inexpensive, and was not an issue for our experiments. Performing injections in younger animals will substantially reduce the amount of virus required. Additionally, a stronger promoter such as CAG could be used to reduce the required amount of virus[23]. Unlike the Synapsin promoter, which is commonly used in neuroscience, CAG would not limit expression to neurons, and despite the strong tropism of PHP.eB AAV to neurons, would result in some glial cells expressing GCaMP. This could be prevented if a FLEX/DIO construct is used, but this would introduce the need for transgenic lines or an additional Cre-expressing virus. An additional caveat concerning PHP.eB and AAV9 serotype AAVs is the clear bias in expression for cortical layer 5, striatum, CA2, and subiculum regions. While the bias towards cortical layer 5 might be explained by the large cell volumes and thus higher capacity for transgene production, this does not appear to be a common feature for the preferred brain areas. An alternative explanation might be that differences in vascularization are determinants of expression levels. However, this is at least unlikely for CA2 of the hippocampus which exhibits strikingly strong expression compared to neighboring hippocampal areas with no large differences in vascularization[24,25]. Notably, the bias for layer 5 neurons was not as clear for ribosome-tethered versions of jGCaMP8. Because transfection efficiency is mainly decided by serotype, this could indicate that it could be a result of signal masking by neuropil. If the mechanisms behind expression differences are identified and reduced in future iterations of synthetic AAV serotypes, fewer viruses may be required to achieve sufficient expression in the upper layers of the cortex for in vivo imaging. Future versions of synthetic AAV serotypes could also, potentially, deliver the virus more efficiently through the BBB to the brain, reducing the required volume of virus.

The need for high brightness of the GECI for systemic administration also caused neuropil contamination of the signals. We therefore made use of two soma-targeting strategies that restricted expression to the cell somata. We show that EE-RR soma-targeting led to the highly stable expression, which was already visible after two weeks, while ribosome-tethering reduced the brightness to such an extent that we did not observe cells during in vivo imaging until 4–6 weeks after injection. Nonetheless, imaging could be performed using reasonable laser power 6 weeks after injection, and the ribosome-tethering led to highly selective although somewhat limited expression in the cell soma. For intracerebral injections, ribosome tethering showed a very high density of cell labeling, but again the expression was slow. We therefore introduced a version with a modified linker region (RiboL1-jGCaMP8) that greatly improved this feature. Using RiboL1-jGCaMP8 we could initiate imaging just one week after intracerebral virus injection, with no apparent drawbacks such as overexpression over extended time periods. In general, ribosome-targeted GECI expression leads to improved signal-to-noise and the possibility to detect activity from a higher number of cells as their activity is not masked by neuropil activity. We also observed that automatic cell detection in Suite2p was more accurate and required smaller data sets from recordings with the ribosome-tethered GECIs.

While reducing neuropil signal is generally favorable when recording from somata, it may in some cases be required for the acquisition of reliable data; a recent paper demonstrates that the signal recorded in striatal fiber photometry or one-photon experiments is mostly produced by neural processes, not striatal cell bodies[26]. Eliminating neuropil signal with a strictly soma-targeted GECI could potentially circumvent this issue, allowing for reliable recordings of somatic $Ca^{2+}$ transients in striatum, or similar brain regions with dense neuropil.

A major challenge when using transgenic animal models to express GECIs is the need for driver lines with general promoters preventing the use of other transgenics. Moreover, co-expression of several viruses in the same brain region, at least in our hands, often proves difficult. In contrast, we show that RO-injected PHP.eB virus is compatible with transgenic lines and co-expression of another virus to obtain cell-specific expression of e.g., a chemogenetic receptor to manipulate their activity or labeling a specific cell population.

In summary, we present a suite of viral vectors for use with both systemic and intracerebral administration that show remarkably high performance and sustainable expression over longer periods. Due to the simplicity of the methods, high experimental flexibility, low-cost and high performance, we believe that these GECI constructs are promising candidates to replace, or complement, transgenic animal models for GECI expression. Our results show that jGCaMP8 and EE-RR-jGCaMP8 are highly suitable for systemic delivery and give rise to brain-wide expression within two weeks and remain stable over months. Finally, the ribosome-tethered jGCaMP8 shows unprecedented labeling density and signal-to-noise that is highly suitable for intracerebral virus injections.

## Methods
### GECI plasmids
All plasmids were transformed into NEB Stable (NEB) competent cells for amplification, and purified using the Zymopure II maxiprep kit (Zymo Research). To obtain Soma-targeted (EE-RR) jGCaMP8, pAAV-Syn-Soma-jGCaMP7 was digested with HpaI and EcoRI to isolate the linker and Soma-tag. The fragment was then ligated into AAV-hSyn-jGCaMP8s, m and f, which was previously digested using the same restriction enzymes. pAAV-Syn-Soma-jGCaMP7[18] was a gift from Edward Boyden (http://n2t.net/addgene:158759;RRID:Addgene_158759). AAV-syn-jGCaMP8s-WPRE[3] was a gift from GENIE Project (http://n2t.net/addgene:162374; RRID:Addgene_162374), as well as jGCaMP8f (Addgene:162376), jGCaMP8m (Addgene:162376) and jGCaMP7c (Addgene: 105320). To obtain ribo-tagged jGCaMP8, pyc126m (Ribo-GCaMP6m) was digested with HpaI and EcoRI to isolate the linker and Ribo-tag. The fragment was then ligated into AAV-syn-jGCaMP8s, m and f, previously linearized using the same restriction enzymes. pycm126[17] was a gift from Jennifer Garrison & Zachary Knight (http://n2t.net/addgene:158777;RRID:Addgene_158777). To obtain Synapsin promoter expressed jREX-GECO1, the jREX-GECO1 coding sequence was cut from CMV-jREX-GECO1 using BamHI and EcoRI, and inserted into a pAAV-Syn-Chr2 plasmid, which was digested with the same restriction enzymes, removing the coding sequence of Chr2 and replacing it with jREX-GECO1. jREX-GECO1 expressed under a CMV promoter was a gift from Neurophotonics[27,28]. The hSyn plasmid, pAAV-Syn_ChR2(H134R)-GFP[29] was a gift from Edward Boyden (http://n2t.net/addgene:58880;RRID:Addgene_58880). To obtain

Ribo-jGCaMP8 with modified linkers, three different linkers were synthesized (Table S1, GeneArt Invitrogen, codon optimized) and inserted into jGCaMP8s, m and f plasmids using HpaI and AccI (NEB). The first linker, RiboL1, was adapted from the SomajGCaMP7f plasmid. The second and third linkers were variations of rigid helical linkers[30], RiboL2: LEA(EAAAK)4ALE, and RiboL3: LEA(EAAAK)4ALEA(EAAAK)4ALE. To obtain axon-targeted jREX-GECO1 and jGCaMP8, a fragment (Table S1) encoding the 20AA axon targeting motif from Broussard et al.[20] was synthesized (Invitrogen, GeneArt) and ligated into jREX-GECO1 and jGCaMP8s and m plasmids. FLEX-RiboL1-jGCaMP8s/f/m plasmids were generated by PCR cloning; briefly, a 5' NheI restriction site was introduced via PCR amplification, and the resulting amplicon containing RiboL1-jGCaMP8 was linearized using Nhel/Ascl, and ligated to a FLEX backbone (#44362) linearized with the same enzymes. Additional plasmids, GCaMP6f, somaGCaMP6f, jGCaMP7f, jGCaMP7s and mNeonGreen were acquired from addgene and were not modified in this paper. pAAV.Syn.GCaMP6f.WPRE.SV40[31] was a gift from Douglas Kim & GENIE Project (http://n2t.net/addgene:100837;RRID:Addgene_100837). pGP-AAV-Syn-jGCaMP7f-WPRE was a gift from Douglas Kim & GENIE Project (http://n2t.net/addgene:104488; RRID:Addgene_104488). pGP-AAV-Syn-jGCaMP7s-WPRE[2] was a gift from Douglas Kim & GENIE Project (http://n2t.net/addgene:104487; RRID:Addgene_104487). pAAV-CAG-mNeonGreen[14] was a gift from Viviana Gradinaru (http://n2t.net/addgene:99134; RRID:Addgene_99134). Plasmids for AAV packaging were acquired from Addgene and Penn Vector core, which is now available from Addgene. Only PHP.eB serotype viruses were used in this paper, except for the cre-dependent DREADD-mCherry, pAAV-hSyn-DIO-hM4D(Gi)-mCherry[32] which was a gift from Bryan Roth (Addgene viral prep # 44362-AAV5;http://n2t.net/addgene:44362;RRID:Addgene_44362). The PHP.eB serotype plasmid, pUCmini-iCAP-PHP.eB[14] was a gift from Viviana Gradinaru (http://n2t.net/addgene:103005;RRID:Addgene_103005). The DeltaF6 helper plasmid, pAdDeltaF6, was a gift from James M. Wilson (http://n2t.net/addgene:112867;RRID:Addgene_112867).

**AAV production.** Viral vectors were produced according to the protocol developed by Challis et al.[33]. Briefly, AAV HEK293T cells (Agilent) were cultured in DMEM with 4.5 g/L glucose & L-Glutamine (Lonza), 10% FBS (Sigma) and 1% PenStrep (Sigma), in a 37 °C humidified incubator. The cells were thawed fresh and split at ~80% confluency until four 182.5 cm² flasks were obtained for each viral prep. The cells were transfected at 80% confluency and the media was exchanged for fresh media directly before transfection. The cells were triple transfected with dF6 helper plasmid and PHP.eB serotype plasmid. Polyethylenimine (PEI), linear, molecular weight (MW) 25,000 (Polysciences, cat. no. 23966-1) was used as the transfection reagent. Media was harvested three days after transfection and kept at 4 °C, and media with cells was harvested five days after transfection and combined with the first media harvest. After 30 min centrifugation at 4000 × g, the cell pellet was incubated with SAN enzyme (Arctic enzymes) for 1 h. The supernatant was mixed 1:5 with PEG and incubated for 2 h on ice, then centrifuged at 4000 × g for 30 min to obtain a PEG pellet containing the virus. The PEG pellet was dissolved in SAN buffer and combined with the SAN cell pellet for incubation at 37 °C for 30 min. To purify the AAV particles, the suspension was centrifuged at 3000 × g for 15 min. The supernatant was loaded on the top layer of an Optiseal tube with gradients consisting of 15%, 25%, 40 and 60% iodixanol (Optiprep). Ultracentrifugation was performed for 2.5 h at 18 °C at 350,000 × g in a type 70 Ti rotor. The interface between the 60 and 40% gradient was extracted along with the 40% layer, avoiding the protein layer on top of the 40% layer. The viral solution was filtered through a Millex- 33 mm PES filter and transferred to an Amicon Ultra-15 centrifugal filter device (100-kDa molecular weight cutoff, Millipore) for buffer exchange. A total of four washes with 13 ml

DPBS were performed at 3000 × g before concentration to a volume of ~500–750 µL. Viral solutions were sterilized using a 13 mm PES syringe filter 0.2 µm (Nalgene), and stored in sterile, low-bind screw-cap vials at 4 °C.

Viral titers were determined using qPCR with primers targeting AAV2 ITR sites[34] (Table S1), following the "AAV Titration by qPCR Using SYBR Green Technology" protocol by Addgene[12]. Briefly, 5 µL of viral sample was added to 39 µL ultrapure H2O, 5 µL 10× DNase buffer, 1 µL DNase, and incubated at 37 °C for 30 min to eliminate all DNA not packaged into AAV capsids. 5 µL of the DNase-treated sample was added to a reaction mix consisting of 10 µL 2× SYBR master mix, 0.15 µL of each primer (100 µM) and 4.7 µL nuclease free H2O. Cycling conditions for the qPCR program were: 98 °C 3 min/98 °C 15 s/58 °C 30 s/ read plate/ repeat 39× from step 3/melt curve.

In addition to the constructs tested in the manuscript, we also made modified versions of them (e.g. FLEX- versions and jGCaMP8f). All plasmids are deposited to Addgene.

**Experimental animals.** All work with experimental animals was performed at the animal facility at the Department of Biosciences, Oslo, Norway, in agreement with guidelines for work with laboratory animals described by the European union (directive 2010/63/EU) and the Norwegian Animal Welfare Act from 2010. The experiments were approved by the National Animal Research Authority of Norway (Mattilsynet, FOTS #14680 and 29491).

Four weeks old male C57/BL6j mice were purchased from Janvier Labs, and housed in GM500 IVC cages in groups of four. After an acclimation period of two weeks, the animals were split into two mice per cage prior to virus injections. One week after injections, the mice were housed individually, and remained single-housed for the duration of the experimental period. The housing room had a 12/12 h light cycle, with lights off at noon. In the light phase, light intensity in the room was 215 lux, and in the cages varied from 20–60 lux, depending on position in the rack. The ambient temperature in the housing room was kept at 21 ± 1 °C, with 25–30% humidity. All experiments were performed in the dark phase. For enrichment purposes, each cage had a running wheel and large amounts of nesting material, and the mice had *ad libitum* access to food and water.

Prior to starting the imaging experiments, the mice were habituated to head-fixation and the running wheel. Each animal was head-fixed for 5–10 min on three to five consecutive days and given Ensure milkshake by a syringe as positive reinforcement. If the mouse at any point showed clear signs of discomfort, they were placed back in their home cage and reintroduced to the apparatus later in the same day.

**Retro-orbital injections.** Pairs of mice were randomly assigned to a viral vector. The mice were placed in an induction chamber and briefly anesthetized by isoflurane, before they were transferred to a mask with 1–2% isoflurane delivered. An eye drop of local anesthetic (oxybuprocaine 4 mg/mL, Bausch Health), was applied to the right eye, and one minute later 100–150 µL of virus (virus titers are listed in Tables 1 and 2) injected into the retro-orbital sinus using a U100 insulin syringe (BD micro-fine 0.3 mL, 30 gauge needle). The volume was determined based on the animal's weight[16]. The surface of the eye was flushed with saline and cleaned with a cotton tip. The mice were then placed back in the home cage and monitored for 10–15 min, before they were returned to the housing room. All animals fully recovered within minutes. In one single mouse, we observed eye damage to the injected side after one week. It is not clear whether this resulted from the injection or resulting from the high incidence of eye abnormalities in the c57bl6 mouse strain[35]. Each viral vector was tested in at least two mice with RO injections and one mouse with intracerebral injection. The GECIs that were deemed bright enough for use with RO injections were further tested in at least three additional mice with both injection techniques.

**Table 2 | Soma and axon-targeted GECI constructs prepared in this work**

| GECI | Titer (VG/ml) | Detectable at 50 mW (RO injection) | Addgene plasmid # |
|---|---|---|---|
| EE-RR-jGCaMP8m (soma) | 1.93E + 13 | Yes | 169257 |
| EE-RR-jGCaMP8s (soma) | 1.73E + 13 | Yes | 169256 |
| EE-RR-jGCaMP8f (soma) | NA | Not tested | 169258 |
| Ribo-jGCaMP8m | 3.27E + 13 | No | 167574 |
| Ribo-jGCaMP8s | 2.80E + 13 | Yes | 167572 |
| Ribo-jGCaMP8f | NA | Not tested | 167573 |
| RiboL1-jGCaMP8s | 1.17E + 13 | Yes | 169247 |
| RiboL1-jGCaMP8m | 3.18E12 | No | 169248 |
| RiboL1-jGCaMP8f | NA | Not tested | 169249 |
| RiboL1-jGCaMP7c | NA | Not tested | 192619 |
| FLEX-RiboL1-jGCaMP8s | NA | Not tested | 192616 |
| FLEX-RiboL1-jGCaMP8m | NA | Not tested | 192617 |
| FLEX-RiboL1-jGCaMP8f | NA | Not tested | 192618 |
| Axon-jGCaMP8m | NA | Not tested | 172719 |
| Axon-jGCaMP8s | NA | Not tested | 172720 |
| Axon-jGCaMP8f | NA | Not tested | 192615 |
| Axon-jREX-GECO1 | 6,66E + 12 | Not tested | 172714 |

**Surgical procedures.** The mice were anesthetized by an intraperitoneal injection of a ketamine/xylazine cocktail (Ketamine 12.5 mg/kg, Pfizer; xylazine 5 mg/kg, Bayer Animal Health GmbH). The top of the head was shaved, and the animals placed on a heating pad in a stereotaxic frame with a mouse adapter (Model 926, David Kopf Instruments). The eyes were covered with white vaseline to prevent drying and to protect them from light. Dexamethasone (5 mg/kg, MSD Animal Health) was delivered via intramuscular injection to prevent edema, and local anesthetic bupivacaine (Aspen) injected in the skin of the scalp. In a subset of animals, the mice were anesthetized by isoflurane (3.5% induction, 1–1.5% maintenance) and additionally injected subcutaneously with buprenorphine (0.05 mg/kg, Indivior Ltd) for analgesia. The skin was cleaned with 70% ethanol and chlorhexidine, and a small piece of skin covering the top of the skull was cut away. The periosteum and other membranes were removed using fine forceps and cotton swabs, and the surface of the skull slightly scored with a scalpel. A custom titanium head post was attached using a few drops of cyanoacrylate and secured using VetBond (3 M) and C&B Metabond (Parkell). A 3.0 mm craniotomy was made using a Perfecta 300 handheld drill (W&H) with a 0.5 mm drill bit (Hager & Meisinger GmbH), centered over primary visual cortex (center coordinates were 2.5 mm ML and 1 mm AP relative to lambda). Custom cranial windows were made by attaching a 3.0 mm diameter round glass (Tower Optical) with 0.45 mm thickness to a 5.0 mm diameter glass (Warner Instruments, USA) with 100 μm thickness using Norland Optical adhesive (Thorlabs GmbH, Germany) under UV light. The cranial window was implanted and secured with C&B Metabond, and a 3D-printed light shield was attached to the head post with black dental acrylic[8]. At the end of the procedures, the mice were injected subcutaneously with 0.3 mL 0.9% saline, meloxicam (5 mg/kg, Boehringer Ingelheim Vet-Medica GmbH), and Antisedan (0.0012 mg/kg, Orion Pharma) (the latter only applies to the mice injected with ketamine/xylazine for anesthesia). Injections of meloxicam were repeated for three days.

In a subset of mice, bone growth partially obscured the view through the cranial window over the course of the experimental period. In these cases, the animal was anesthetized by isoflurane, the window removed to clear away any bone growth and other debris, and a new cranial window implanted[8]. The procedure was performed one week prior to imaging to allow the tissue to recover from potential damage during bone removal.

**Intracerebral virus injections.** Glass capillaries (OD 1.14 mm: ID 0.53 mm) were pulled and beveled at a 40-degree angle[36], and mounted in a NanoJect 3 (Drummond Scientific, USA). The pipette was front-loaded with the virus solution and 150nL injected at a depth of 350–500 μm below the dura, in 5 nL steps. After the last injection, the pipette was left in the tissue for five minutes before retraction and loading of a new pipette. Two to three different constructs were injected per animal, spaced at least 700 μm apart. After the final injection, the exposed brain surface was cleaned with saline, and a cranial window implanted as described above.

**Co-expression of Axon-jREX-GECO1 and soma-targeted GECIs.** For co-expression of Axon-jREX-GECO1 and RiboL1-jGCaMP8, surgical procedures were conducted as described above. After attachment of the head post, a small craniotomy was made above the dLGN, and 150 nL of virus slowly injected in 5 nL steps over 5 min (injection coordinates relative to bregma were 2.1 mm posterior, 2.3 mm lateral, and 2.5 mm below the dura). A larger craniotomy was then made over V1, RiboL1-jGCaMP8s was injected and a cranial window implanted as described above.

**Wide-field imaging.** Wide-field imaging was used to monitor the expression levels of the Ca$^{2+}$ sensors and quality of the cranial windows. The mice were head-fixed on a custom 3D-printed running wheel using optical posts that were mounted to the optical table, holding clamps (Standa) and modified ball-joints (Thorlabs GmbH) allowing for adjustments in AP elevation. Single images were acquired by a Canon EOS 4000D camera through a ×5 Mitutoyo long working distance objective (0.14 NA) in an Olympus BX-2 microscope. The light source was a xenon arc Lambda XL lamp (Sutter Instruments) with 480/545 nm and 560/635 nm filters (#39002 and #39010, Chroma). All animals were imaged using two sets of parameters at each time-point, with exposure times of 600 and 2000 ms, and ISO of 100 and 400, respectively. The mice ran freely in darkness during imaging. In addition, wide-field videos were captured at 25 Hz during both spontaneous activity (in darkness) and with visual stimulation.

**Two-photon imaging.** For in vivo two-photon imaging we used a resonant-galvo Movable Objective Microscope (MOM, Sutter Instruments) with a MaiTai DeepSee laser (SpectraPhysics) set to a wavelength of 920 nm. The main objective used for screening was a ×10 objective (TL-10 × 2P, 0.5 NA, 7.77 mm working distance, Thorlabs), giving a field of view of approximately 1665 × 1390 μm. In mice with successful GCaMP imaging, we also imaged at lower depths (200–500 μm) using a Nikon ×16 objective (NA 0.8), giving a field of view of approximately 1050 × 890 μm, or an Olympus ×20 objective (NA 1.0) giving a field of view of ~840 × 700 μm. The laser was controlled by a pockel's cell (302 RM, Conoptics), and fluorescence was detected through Chroma bandpass filters (HQ535-50-2p and HQ610-75-2p, Chroma) by PMTs (H10770PA-40, Hamamatsu). Images were acquired at 30.9 Hz using MCS software (Sutter Instruments). Output power at the front aperture of the objective was measured prior to each imaging session with a FieldMate power meter (Coherent) and set to 45 mW, unless mentioned otherwise. The microscope was tilted to an angle of 6–12 degrees during imaging to match the surface of the brain, in addition to the 4-6 degree forward tilt made by the head-fixing apparatus.

**Table 3 | List of antibodies used for post-mortem histology**

| Antibody | Supplier | Dilution used | RRID | Cat # |
|---|---|---|---|---|
| Chicken anti-GFP | Invitrogen | 1:1000 | AB_2534023 | A10262 |
| Rabbit anti-NeuN | Abcam | 1:100 | AB_2532109 | Ab177487 |
| Goat anti-TdTomato | Sicgen | 1:2000 | AB_2722750 | AB8181 |
| Rabbit anti-parvalbumin | Swant | 1:2000 | AB_2631173 | PV27 |
| Goat anti-Iba1 | Invitrogen | 1:500 | AB_10982846 | PA5-18039 |
| Donkey anti-goat IgG, CF 568 conj. | Biotium | 1:1000 | AB_10854239 | 20106-1 |
| Goat anti-chicken IgG, AF 488 conj, | Invitrogen | 1:1000 | AB_142924 | A-11039 |
| Goat anti-rabbit IgG, AF 647 conj, | Invitrogen | 1:1000 | AB_2535813 | A-21245 |
| Chicken anti-rabbit IgG, AF 488 conj. | Invitrogen | 1:1000 | AB_2535859 | A-21441 |
| Donkey anti-chicken IgY, CF 488 conj. | Sigma Aldrich | 1:1000 | AB_2721061 | SAB4600031 |
| Donkey anti-goat IgG, AF 647 conj. | Invitrogen | 1:1000 | AB_2535864 | A-21447 |

**Co-expression of GCaMP and cell-specific DREADDs.** PV-Cre mice (Jackson Laboratories, strain #017320) were injected in the retro-orbital sinus with Soma-jGCaMP8s as described above. Two weeks later, 150nL of pAAV5-hSyn-DIO-hM4D(Gi)-mCherry (#44362, Addgene) was injected into the cortex, and a cranial window implanted.

**Visual stimulus.** Sinusoidal drifting gratings were generated using the open-source Python software PsychoPy[37], and synchronized with two-photon imaging through a parallel port and PCIe-6321 data acquisition board (National Instruments). We used drifting gratings of three orientations (0, 135, and 270 degrees) with a spatial frequency of four cycles per degree and temporal frequency of 2 Hz. Stimulus was shown for 3 seconds, interleaved with an 8 second gray screen period.

**Two-photon imaging analysis.** Motion-correction and automatic detection of regions of interest (ROIs) was performed using suite2p[38]. The data were then manually curated and analyzed using custom Python scripts. To calculate the relative fluorescence $\Delta F/F0$, traces were first corrected for neuropil using $Fc = (F - 0.7 * Fneu + 0.7 * Fneu\_median)$, where $F$ is the raw fluorescence, Fneu is the neuropil signal defined by suite2p as the fluorescence in a circle around the ROI, and Fneu_median is the median of that signal. $\Delta F/F0$ was then defined as $(Fc - F0)/F0$, where $F0$ is the median of the neuropil-corrected trace $Fc$. The soma/neuropil correlation was defined as the Pearson Correlation Coefficient between the raw fluorescence trace F and the neuropil trace Fneu for each ROI. The soma/neuropil baseline ratio was determined as the ratio between the median of the raw fluorescence trace $F$ and the median of the neuropil trace Fneu for each ROI over the entire recording time. The baseline fluorescence of an ROI was defined as the median of the raw fluorescence $F$. Stimulus-evoked event rates were calculated based on the activity during the 3-second stimulus presentation, and spontaneous event rates were calculated from the activity during the 8-second inter-trial interval for the same ROIs.

**Soma-axon pair correlation.** To determine the correlations between soma-axon pairs, we defined a threshold at 2 * std of the $\Delta F/F0$ trace of the soma to identify spikes. When at least two subsequent frames exceeded the threshold, a window from 324 ms before to 1942 ms after the spike was defined. For each of these windows, the Pearson Correlation Coefficient was determined between the soma $\Delta F/F0$ trace and the corresponding axon $\Delta F/F0$ trace. For proof-of-principle purposes, we used RiboL1-jGCaMP8s signal from one representative soma and Axon-jRex-GECO signals from 7 representative boutons (5 with clear spatial separation from the soma and 2 in close proximity to the soma).

**FOV fluorescence intensity analysis.** We used average intensity projections of 500 frames, acquired six weeks after virus injection, to measure the signal intensity across the FOV. The intensity as measured at three y coordinates evenly distributed across the FOV using ImageJ and normalized by dividing all values to the single highest pixel value within each measurement set. The data is down sampled from 512 to 330 measurement points, and the average values from the three individual measurements per mouse were used for data presentation.

**ROI per field of view analysis.** We used average intensity projections of 250 frames, acquired six weeks after virus injection, to measure the number of ROIs per FOV. One single FOV per mouse was used for the analysis, and the segmentation was performed by using CellPose[39]. The images for the analysis were acquired using identical settings and a FOV size of $840 \times 700 \, \mu m$.

**Wide-field imaging analysis.** To measure changes in fluorescence over time in wide-field imaging videos, we used ImageJ (Fiji). Videos were spatially down-sampled, and regions of interest (ROIs) were selected in the center of the cranial window. Changes in relative fluorescence were calculated by $(F-F_0/F_0)$, where the baseline fluorescence ($F_0$) was defined as the mean fluorescence across all frames from "spontaneous" and "visual stimulus" runs in the entire cranial window. Calcium signal traces were obtained from the average fluorescence intensity in an ~200 $\mu m$ diameter circular area.

**Histology.** Six weeks after virus injection, all animals were deeply anesthetized by an intraperitoneal injection of Euthasol (pentobarbital sodium 100 mg/kg, Le Vet) and intracardially perfused with PBS followed by 4% paraformaldehyde in PBS. Brains were dissected out and postfixed for 24 h followed by cryoprotection in 30% sucrose in PBS for 48 h. 40 $\mu m$ coronal sections were cut with a cryostat (Leica). All sections were stained free-floating on constant agitation. The sections were rinsed three times in PBS followed by blocking in 2% bovine serum in 0.3 % Triton X-100 in PBS for 1 h before incubation with the primary antibody in blocking solution overnight (all antibodies used are listed in Table 3). After rinsing, sections were incubated with secondary antibodies in PBS for 1 h. Sections were then mounted on Superfrost Plus adhesion slides and dried for 2 h. After rinsing in dH$_2$O and additional drying for 1 h, sections were coverslipped with mounting medium (Ibidi). Tile scans were acquired with 20% overlap on an Andor Dragonfly spinning-disc microscope with a motorized platform and stitched using Fusion software (Bitplane). The Andor Dragonfly was built on a Nikon TiE inverted microscope equipped with a Nikon PlanApo ×10/0.45 NA objective. High-magnification images were acquired on the same microscope using a Nikon CF Apo ×60/1.4 NA objective.

To estimate the number of activated microglia per area, we used sections stained for Iba1, in addition to the neuronal marker NeuN and GFP (to verify the expression of the GECI). A region containing the

primary visual cortex was selected, and the number of Iba1-positive cells was manually counted using ImageJ.

**Reporting summary**

Further information on research design is available in the Nature Portfolio Reporting Summary linked to this article.

## Data availability

All plasmids used have been uploaded to Addgene, and antibodies used are listed with their corresponding RRID in Table 3. Original raw data from imaging experiments will be available in a public repository with time, and are available upon request to the authors. Source data are provided with this paper.

## Code availability

Custom code used for analysis is available at https://github.com/frederikrogge/calcium-sensor-analysis.

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

## Acknowledgements

We thank Dr. Ane Charlotte Christensen for assistance with the establishment of the RO injection technique, and for feedback on the manuscript. We also thank the late Paul Johannes Helm for assistance with assembling the two-photon microscope and for many enjoyable discussions about microscopy, physics, and biology. Finally, we thank the Instrumentation Lab at the Department of Bioscience. Imaging of histological sections was performed at the NorMIC microscopy platform at the Department of Bioscience, University of Oslo.

## Author contributions

S.G., I.N., G.H.V., K.K.L., and M.F. designed the study. S.G. and G.H.V. performed plasmid design and assembly, and virus production, with assistance from V.B., I.N., and K.K.L. performed surgeries and in vivo imaging. S.G. and I.N. performed immunohistochemistry and post-mortem imaging. F.S.R., S.G., I.N., and K.K.L. analyzed data. S.G., I.N., and K.K.L. wrote the manuscript with inputs from all co-authors. All authors have read the final version of the manuscript.

## Competing interests

The authors declare no competing interests.
