## [Peer Review File · Nature Communications]

An updated suite of viral vectors for in vivo calcium imaging using intracerebral and retro-orbital injections in male mice.REVIEWER COMMENTS

Reviewer #1 (Remarks to the Author):

In this manuscript, “An updated suite of viral vectors for in vivo calcium imaging using local and retro-orbital injections,” the authors describe the use of retro-orbital injections of AAV PHP.eB GCaMPs for systemic delivery of genetically-encoded calcium indicators. This method allows for cerebral expression without the need to develop transgenic animals or perform intracranial injections. They further optimize GCaMP8 soma localization using a ribosome tethering strategy and adjusted linker sequence to improve somatic localization and expression time. The data included here represent a large undertaking, to characterize retro-orbital delivery of several different GCaMP iterations and expression of novel ribosome-tethered GCaMP8 variants. These will certainly prove useful to the field, describing systemic delivery of GCaMP expression as well as a more efficient soma-localized GCaMP8. However, a number of concerns, foremost among them limited quantification and experimental n, should be addressed as outlined below for reconsideration of publication.

Major issues:

1. Throughout the paper, there is a lack of quantification of the results as described, as well as very limited sample sizes. For example, quantification of GCaMP expression across cortical layers (Figures 1 & 2), GCaMP “brightness” (Tables 1 & 2), and number of cells identified per imaging field with intracranial vs retro-orbital injections and GCaMP8 iterations (Figures 1, 2, 3, 4) would provide a more useful dataset for those in the field to evaluate the suitability of these tools for use in different experimental paradigms. Throughout the paper, and especially for quantification, multiple animals per condition and multiple imaging sessions per animal should be demonstrated as viral expression across animals could be variable. Further, the authors present a large body of work describing retro-orbital delivery and optimized soma localization of many GCaMPs. In order to avoid using a large number of additional animals for revised work, we would suggest that the authors focus such efforts on a subset of their most promising or interesting conditions (e.g. retro-orbital vs intracranial delivery of GCaMP8s and RiboL1-GCaMP8s) as many viral conditions are currently included.
2. A quantitative assessment of expression stability, cell health, and tissue health (i.e. microglial or astrocytic activation) should be performed for at least a subset of viral delivery methods and soma-localized GCaMP8 expression.
3. Supplemental figures and tables were not received for review.

Minor issues:

1. We would suggest incorporating more detailed experimental descriptions into the figures and/or text, being sure to include at what timepoint imaging was performed, number of animals, and number of imaging sessions where pertinent.
2. The reason for the transition from retro-orbital vs intracranial viral delivery to development of a ribosome-tethered GCaMP8 is unclear in the text. A quantification of neuropil vs soma GCaMP8 signal (or SNR) would be helpful for the reader in justifying the need for a soma-localized GCaMP8.
3. Figure 1:

a. Figure 1C should include higher magnification images across timepoints, for assessment of cell morphology and GCaMP aggregation. As noted above, it would be useful to quantify such metrics for a subset of conditions. High magnification post-mortem examples would be helpful as well.

b. In Figure 1D, we suggest quantification of expression across cortical layers for both retro-orbital and intracranial viral injections.

4. Figure 2:

a. The time point after injection is unclear here. Again, expression should be quantified across cortical layers (which should also be labeled in the figure).

b. Again, higher magnification 2-photon images would be helpful. Quantification of cells identified and SNR might provide justification for development of soma-localized GCaMP8.

5. Tables 1 & 2:

a. "Brightness" should be quantified and the authors must explain how brightness levels were decided.

b. The authors might comment on the potential variability in expression characteristics based on viral titers (different viruses may require different viral particle numbers).

6. Figure 3:

a. We suggest first showing expression of ribosome-tethered GCaMP8 using the more established technique of intracranial injection, followed by retro-orbital injections.

b. The time point post-injection is unclear here.

c. In Figure 3B – it may be useful to quantify metrics across GCaMPs tested (such as number of cells identified per FOV, kinetics, could also include neuropil quantification here). Otherwise, the representative image and dF/F traces are of limited utility.

d. Although the authors state in text that Ribo-GCaMP8s and RiboL1-GCaMP8s retro-orbital delivery do not result in sufficient expression, Ribo-GCaMP8s retro-orbital expression is shown in Figure 3A. Given improved expression time, we suggest including a panel of retro-orbital injection with RiboL1-GCaMP8s (even if dim or higher imaging power required).

7. Figure 4: quantification of cells identified per field would be useful.

8. Figure 5:

a. The authors might include quantification of soma alone and neuropil alone here, particularly to show (anticipated) reduction in neuropil signal with ribosome-tethered GCaMP8.

b. Higher n (cells, animals) should be used, and number of sessions imaged should be noted.

9. Figure 6:

a. This figure provides limited additional novelty in the scope of this paper. We would suggest including more motivation as to the novelty and/or importance of this combination of techniques or moving this figure to the supplement.

10. Figure 7:

a. If commenting on correlation of axonal and somatic signals, authors should perform at least a correlation analysis here.

Reviewer #2 (Remarks to the Author):

The data presented in this manuscript represent an extensive analysis of the performance of various calcium sensors expressed through AAV systemic or local delivery. The comparison of various versions of GCaMP represents interesting and relevant results to be shared with the community and I greatly appreciate that the authors have included negative results for most of versions of the sensor. The timelines of expression are interesting as well.

The development of new ribo-constructs as an attempt to promote soma targeting is interesting and promising but unfortunately only showed significant improvements compared to the existing linker version when injected locally.

Experimental procedures and methodologies are appropriately described.

It would have been nice to see the performance of JReXGeco in RO injections. Considering that numerous specific promoters/enhancers are emerging, the use of systemic injections for AAVs could lead to targeted expression... The authors did discuss issues of the viral load required for PHP.eB IV injections but this issue could become less significant with even better versions of these BBB crossing serotypes being generated.

It would have been relevant for the authors to discuss other strategies/methodologies used to allow AAVs to cross BBB such as FUS which is documented to be compatible with AAV transduction.

Overall, despite the quality of the data and the general interest of this analysis, the novelty is lacking. Specifically, a paper by Michelson et al. (*Neurophotonics* 6(2), 025014 (Apr–Jun 2019)) has already reported that GCaMP6s delivered through an AAV PHP.eB is expressed at levels comparable, if not better than the equivalent transgenic mice. Even if their study was done using 1p imaging, this difference is not of major significance.

Reviewer #3 (Remarks to the Author):

There are multiple advantages of viral vectors for driving expression of GECIs as opposed to using transgenic animals. The engineered capsid PHP.eB enables systemic delivery of AAVs which then drive expression in brain. This paper provides practical guidance for using viral vectors for labeling neurons using both systemic and localized injection. Several different types of GCaMP were screened for their effectiveness when used with the systemic injection and the newer 7 and 8 versions was found to be better expressed as quantified by brightness than GCaMP6. The authors also made new soma-targeted jGCaMP8 variants including one with a novel linker that showed faster expression. Some of the findings in this paper will be useful for experimental neuroscientists in facilitating experiments. However, there is very little quantification of expression rates, metrics of functionality, or assays for artifacts such as altered firing. This makes the paper limited in its utility and decreases the confidence in the assessment

of the labeling strategies.

Major points

Qualitative assessment of brightness of the labeling, which depending on how it is measured, may or may not include the intensity during firing, is used as one of the primary metrics. It would be important to characterize whether this a measure of activity or baseline fluorescence or some combination of both. For example, in Figure 2, is brightness integrated over time and during what behavior or anesthesia? It is not clear that brightness in projections is a good sign for GECIs as a darker average background may be useful, while some bright cells are pathological. Only examples of fluorescence dF/F_0 tracers are shown, and there is no comparison across multiple samples.

Some of the data come from very small numbers of animals (Fig. 5 says single animals), so there is no way of assessing variability of the labeling. This is very important information for a paper assessing a scientific resource.

The use of different colors for axonal and soma targeted recording is intriguing. However, the signal from the axons is quite noisy and there is no verification or aggregate quantification. In the example traces look as though the axons have a large increase in fluorescence whenever the soma fires. It is also possible that the axonal signal is bleed through to the axonal channel. The separation in the channels should be confirmed with quantification of the correlation.

Especially for the novel soma-targeted GCaMP, need to evaluate whether this affects the firing behavior of the labeled cell.

Specific points

How does the number of labeled neurons (perhaps measured by the ratio of anti-GFP to NeuN labeling) depend on the GCaMP type (Fig. 2b)? The number of labeled cells looks quite different in Ribo and soma images (Fig. 3). Is this because the firing is different or there are different number of neurons labeled.

Do the really bright cells still have normal firing?

Please also clarify the promoter used in each experiment.

For soma-targeted GCaMP (again especially for the new constructs), please show higher magnification images of the novel soma targeting versions to show where the expression is. Would likely need a higher NA objective than 0.8NA used here. Also need quantification of change in fluorescence from the soma versus the surround, rather than just the base-line brightness.

In widefield imaging, how was potential variations in activity due to different amounts of running or other behaviors accounted for across the mice?

Fig. 1C and D: Please state which GECI was used in the example

Intro: "Overexpression in a subset of cells can lead to intracellular aggregation and eventually cell death." Did you mean that cells are protected when there is overexpression everywhere or that overexpression in any number of cells is problematic?

It not really neuropil "contamination", as the neuropil signal itself can be useful. Perhaps there is better

wording for this as often the signal does reflect true neuronal activity.

Please define MOI in text better and move the speculation about the mechanisms to Discussion. Otherwise, it would be good to measure the viral vector genetic material in the cells.

Do you see GCaMP accumulation in the surrounding tissues such as the in the dura and if so, was it different with the various delivery options?

“In an attempt to improve the brightness, we tested both double and triple injection volumes of RO administered GCaMP6f, but the resulting expression was still too dim to image at reasonable laser power (data not shown).” Showing even supplementary data on this would be useful for the community so other people don’t have to try it.

In Results: “the expression remained stable across weeks (Fig. 1C), with no indication of intracellular aggregation.” It looks like it is increasing in brightness at 6 weeks.

Fig. 2b caption – please also indicate that this is GFP antibody labeling in caption.

Fig. 3b – are the traces shown from particular neurons indicated on the left? Hard to tell from the color code in left panel.

In Results: “local injections of two or more viruses often leads to competition and low co-expression” – Please provide substantiation for this statement or eliminate. There’s lots of anecdotal evidence that this works quite well.

“flexed hM4D DREADD receptor” should this be “floxed”?

In discussion “An additional caveat concerning the PHP.eB and AAV9 is that we find a clear bias in expression for cortical layer 5, striatum, CA2 and subiculum regions.” Please show a quantification of this. The effect is not so clear that I can tell from the images.

Metacam/meloxicam: please use consistent wording

Were imaging sessions awake or anesthetized? Please provide details for either (e.g. training for awake imaging)

RESPONSE TO REVIEWER COMMENTS

We thank the reviewers for their helpful comments and suggestions to our manuscript. We have made changes to the manuscript according to the reviewers' suggestions, and added data from several new experiments that strengthen both the original data sets and insight from the experiments. We believe that the manuscript is substantially improved.

Below are the reviewers' comments with our replies in blue text.

Reviewer #1 (Remarks to the Author):

In this manuscript, "An updated suite of viral vectors for in vivo calcium imaging using local and retro-orbital injections," the authors describe the use of retro-orbital injections of AAV PHP.eB GCaMPs for systemic delivery of genetically-encoded calcium indicators. This method allows for cerebral expression without the need to develop transgenic animals or perform intracranial injections. They further optimize GCaMP8 soma localization using a ribosome tethering strategy and adjusted linker sequence to improve somatic localization and expression time. The data included here represent a large undertaking, to characterize retro-orbital delivery of several different GCaMP iterations and expression of novel ribosome-tethered GCaMP8 variants. These will certainly prove useful to the field, describing systemic delivery of GCaMP expression as well as a more efficient soma-localized GCaMP8. However, a number of concerns, foremost among them limited quantification and experimental n, should be addressed as outlined below for reconsideration of publication.

Major issues:

1. Throughout the paper, there is a lack of quantification of the results as described, as well as very limited sample sizes. For example, quantification of GCaMP expression across cortical layers (Figures 1 & 2), GCaMP "brightness" (Tables 1 & 2), and number of cells identified per imaging field with intracranial vs retro-orbital injections and GCaMP8 iterations (Figures 1, 2, 3, 4) would provide a more useful dataset for those in the field to evaluate the suitability of these tools for use in different experimental paradigms.

We have removed the unclear phrasing around the term "brightness" and replaced it with a simple yes/no categorization of "Detectable at 50mW (RO injections)". This is defined in

the text as whether a single imaging plane using a 16X Nikon objective and 50mW laser power at 920nm gives clear single-cell fluorescence transients.

Brightness quantification in and of itself is not clear-cut, and many former papers use different approaches. In an attempt to present the data as unbiased as possible, we have included analysis on baseline brightness and the correlation between soma and neuropil signals. Moreover, we have quantified the number of identified cells per FOV using average intensity projections, and the fluorescence intensity across the FOVs. These analyses are all added to a new figure panel (Fig 5).

Throughout the paper, and especially for quantification, multiple animals per condition and multiple imaging sessions per animal should be demonstrated as viral expression across animals could be variable. Further, the authors present a large body of work describing retro-orbital delivery and optimized soma localization of many GCaMPs. In order to avoid using a large number of additional animals for revised work, we would suggest that the authors focus such efforts on a subset of their most promising or interesting conditions (e.g. retro-orbital vs intracranial delivery of GCaMP8s and RiboL1-GCaMP8s) as many viral conditions are currently included.

We have performed additional experiments and added data from animals with both injection methods for jGCaMP8s, EE-RR-jGCaMP8s (soma-) and RiboL1-jGCaMP8s, with local injections of GCaMP6f for comparison. The analysis for these GECIs now includes data from 3-6 individual mice for each indicator and injection technique.

2. A quantitative assessment of expression stability, cell health, and tissue health (i.e. microglial or astrocytic activation) should be performed for at least a subset of viral delivery methods and soma-localized GCaMP8 expression.

We have included staining and quantification of astrocytic activation (Fig S5). Moreover, we have expanded on the stability/cell health, and added high-resolution images from 2, 4, 6, 8 and 10 weeks after injection (Fig 4).

3. Supplemental figures and tables were not received for review.

We apologize for this mistake. Supplementary data is now included.

Minor issues:

1. We would suggest incorporating more detailed experimental descriptions into the figures and/or text, being sure to include at what timepoint imaging was performed, number of animals, and number of imaging sessions where pertinent.

We have added more detailed descriptions in the figures and the figure texts.

2. The reason for the transition from retro-orbital vs intracranial viral delivery to development of a ribosome-tethered GCaMP8 is unclear in the text. A quantification of neuropil vs soma GCaMP8 signal (or SNR) would be helpful for the reader in justifying the need for a soma-localized GCaMP8.

The text is now changed to better explain the shift in focus to soma localization. We believe the analysis of soma/neuropil correlation and the number of identified cells per FOV provide a good explanation as to why soma localization is important when using these ultrasensitive GECIs. We have also added a paragraph in the discussion about the importance of strict soma-targeting with these new bright versions of GCaMP for imaging deeper areas of the brain where the neuropil signal completely dominates the soma signals, as per a recent paper (Legaria et al., 2022).

3. Figure 1:

a. Figure 1C should include higher magnification images across timepoints, for assessment of cell morphology and GCaMP aggregation. As noted above, it would be useful to quantify such metrics for a subset of conditions. High magnification post-mortem examples would be helpful as well.

We have changed the example images in Fig 1 to include more time points. As described above, we have added high-resolution images from several GECIs using both injection techniques in a separate panel (Fig 4). Post-mortem histology is included in the Supplementary Data (Fig S2 and S3).

b. In Figure 1D, we suggest quantification of expression across cortical layers for both retro-orbital and intracranial viral injections.

For the 3 RO injected animals that we show example data from in Fig 3, the distributions are relatively clear (see figure below, each dot represents a FOV from one animal). However, we do not have consistent sampling across cell layers with identical sampling depths and acquisition settings, so we have not included this in the manuscript.

4. Figure 2:

- a. The time point after injection is unclear here. Again, expression should be quantified across cortical layers (which should also be labeled in the figure).
- b. Again, higher magnification 2-photon images would be helpful. Quantification of cells identified and SNR might provide justification for development of soma-localized GCaMP8. We have clarified the time points in the legend. As stated above we have included quantification of identified cells. We have also included analysis of the cross-correlation between the neuropil and soma signals, and the soma/neuropil signal intensity relationship. Together, these analyses all indicate a strong improvement in signal quality and detection with ribosome-tethering compared to traditional GCaMP.

5. Tables 1 & 2:

- a. "Brightness" should be quantified and the authors must explain how brightness levels were decided.
- b. The authors might comment on the potential variability in expression characteristics based on viral titers (different viruses may require different viral particle numbers). See note above about the brightness quantification. We have added a brief statement about virus titers and multiplicity of infection in the Discussion.

6. Figure 3:

- a. We suggest first showing expression of ribosome-tethered GCaMP8 using the more established technique of intracranial injection, followed by retro-orbital injections. These are now both included in the same figure panel (Figure 3) and the new version of figure 4.
- b. The time point post-injection is unclear here.

This has been clarified in the legend for all figures.

c. In Figure 3B – it may be useful to quantify metrics across GCaMPs tested (such as number of cells identified per FOV, kinetics, could also include neuropil quantification here). Otherwise, the representative image and dF/F traces are of limited utility.

We have removed figure 3B and added more quantification in figure 5.

d. Although the authors state in text that Ribo-GCaMP8s and RiboL1-GCaMP8s retro-orbital delivery do not result in sufficient expression, Ribo-GCaMP8s retro-orbital expression is shown in Figure 3A. Given improved expression time, we suggest including a panel of retro-orbital injection with RiboL1-GCaMP8s (even if dim or higher imaging power required).

High-resolution images of RO injected RiboL1-jGCaMP8s with expression over time has been added to the new Figure 4.

7. Figure 4: quantification of cells identified per field would be useful.

This has been added for comparing the expression of Ribo-jGCaMP8 with RiboL1-jGCaMP8 in Figure 3, and for GCaMP6f, EE-RR, RiboL1 and regular jGCaMP8 using both injection methods in Figure 5.

8. Figure 5:

a. The authors might include quantification of soma alone and neuropil alone here, particularly to show (anticipated) reduction in neuropil signal with ribosome-tethered GCaMP8

As stated above, we have added analysis of soma-neuropil signal correlations and ratios.

b. Higher n (cells, animals) should be used, and number of sessions imaged should be noted.

We have increased the number of animals to 3-4 per group.

9. Figure 6:

a. This figure provides limited additional novelty in the scope of this paper. We would suggest including more motivation as to the novelty and/or importance of this combination of techniques or moving this figure to the supplement.

We agree and have moved the figure to Supplementary Figure 7. The data showing tdt expression only in PV-Cre mice has been removed from the results altogether, as this has been shown several times by others.

10. Figure 7:

a. If commenting on correlation of axonal and somatic signals, authors should perform at least a correlation analysis here.

We have added analysis of cross-correlation during periods of strong Ca²⁺ transients in the soma (whenever it exceeds 2x the standard deviation of the baseline signal). This indicates that the axonal boutons show both highly correlated and non-correlated signals with the soma.

Reviewer #2 (Remarks to the Author):

The data presented in this manuscript represent an extensive analysis of the performance of various calcium sensors expressed through AAV systemic or local delivery. The comparison of various versions of GCaMP represents interesting and relevant results to be shared with the community and I greatly appreciate that the authors have included negative results for most f versions of the sensor. The timelines of expression are interesting as well.

The development of new ribo-constructs as an attempt to promote soma targeting is interesting and promising but unfortunately only showed significant improvements compared to the existing linker version when injected locally.

Experimental procedures and methodologies are appropriately described.

It would have been nice to see the performance of JRexGeco in RO injections.

We show that jRexGeco expresses well with RO injections. However, previously published work shows that this construct has low sensitivity and slow kinetics. Both jRexGeco and other red GECIs would be interesting to screen. Due to time constraints we mainly prioritized the more sensitive GECIs for RO injections, but have expanded our analysis on the axon-targeted jRex-GECO.

Considering that numerous specific promoters/enhancers are emerging, the use of systemic injections for AAVs could lead to targeted expression... The authors did discuss

issues of the viral load required for PHP.eB IV injections but this issue could become less significant with even better versions of these BBB crossing serotypes being generated.

We appreciate this suggestion from the reviewer and have added a sentence on the potential of future versions of synthetic AAVs to more efficiently deliver virus through the BBB to the brain.

It would have been relevant for the authors to discuss other strategies/methodologies used to allow AAVs to cross BBB such as FUS which is documented to be compatible with AAV transduction.

While FUS mediated AAV transduction is an impressive method, it is out of the scope of this paper. A discussion of other delivery methods, such as n-SIM, has been included.

Overall, despite the quality of the data and the general interest of this analysis, the novelty is lacking. Specifically, a paper by Michelson et al. (Neurophotonics 6(2), 025014 (Apr–Jun 2019)) has already reported that GCaMP6s delivered through an AAV PHP.eB is expressed at levels comparable, if not better than the equivalent transgenic mice. Even if their study was done using 1p imaging, this difference is not of major significance.

The Michelson et al paper only images in widefield 1P, which does not require the same signal intensity as 2PLSM imaging requires. They show that brightness in 1p is similar (or the same), not better, for one out of three lines tested, the other two transgenic lines are much brighter at lower laser powers. The authors also report having to use “relatively high LED power” to image the PHP.eB infected (and the Thy1) mouse. Additionally, they only quantify brightness for a single mouse per line. We believe our manuscript brings considerable novelty.

Reviewer #3 (Remarks to the Author):

There are multiple advantages of viral vectors for driving expression of GECIs as opposed to using transgenic animals. The engineered capsid PHB.eB enables systemic delivery of AAVs which then drive expression in brain. This paper provides practical guidance for using viral vectors for labeling neurons using both systemic and localized injection. Several different types of GCaMP were screened for their effectiveness when used with the systemic injection and the newer 7 and 8 versions was found to be better expressed as quantified by brightness than GCaMP6. The authors also made new soma-targeted

jGCaMP8 variants including one with a novel linker that showed faster expression. Some of the findings in this paper will be useful for experimental neuroscientists in facilitating experiments. However, there is very little quantification of expression rates, metrics of functionality, or assays for artifacts such as altered firing. This makes the paper limited in its utility and decreases the confidence in the assessment of the labeling strategies.

Major points

Qualitative assessment of brightness of the labeling, which depending on how it is measured, may or may not include the intensity during firing, is used as one of the primary metrics. It would be important to characterize whether this a measure of activity or baseline fluorescence or some combination of both. For example, in Figure 2, is brightness integrated over time and during what behavior or anesthesia? It is not clear that brightness in projections is a good sign for GECIs as a darker average background may be useful, while some bright cells are pathological. Only examples of fluorescence dF/F_0 tracers are shown, and there is no comparison across multiple samples.

The experimental conditions have been clarified in the Methods section and in the text in Results section.

As noted in our reply to Reviewer 1, we have added analysis of the correlation between soma and neuropil signals, the ratios of baseline signal between soma and neuropil, and baseline fluorescence.

We have also added two figures to investigate the potential toxicity of longitudinal expression using both injection techniques; in Figure 4 we have tracked the same populations of cells from 2-10 weeks after virus injections using a high-NA 20X objective, and in Figure S6 we have analyzed the number of activated microglia by staining tissue sections for the marker Iba1.

Some of the data come from very small numbers of animals (Fig. 5 says single animals), so there is no way of assessing variability of the labeling. This is very important information for a paper assessing a scientific resource.

We have performed additional experiments using GCaMP6f and the most promising new iterations of GCaMP constructs (jGCaMP8s, EE-RR-jGCaMP8s and RiboL1-jGCaMP8s). For all testing and analysis purposes, the number of animals for all these is now 3-5 per GEI for each injection method.

The use of different colors for axonal and soma targeted recording is intriguing. However, the signal from the axons is quite noisy and there is no verification or aggregate quantification. In the example traces look as though the axons have a large increase in fluorescence whenever the soma fires. It is also possible that the axonal signal is bleed through to the axonal channel. The separation in the channels should be confirmed with quantification of the correlation.

We have added analysis of cross-correlation during periods of strong Ca^{2+} transients in the soma (whenever it exceeds 2x the standard deviation of the baseline signal). This indicates that the axonal boutons show both highly correlated and uncorrelated signals with the soma, indicating that bleed-through is not a major concern.

Especially for the novel soma-targeted GCaMP, need to evaluate whether this affects the firing behavior of the labeled cell.

We have added a figure (S5) showing the responses to visual stimuli from EE-RR and RiboL1 versions of jGCaMP8 expressed through an intracerebral injection. In the additional experiments performed for the revision, we have expressed RiboL1-jGCaMP8s in the same cranial window as GCaMP6f for direct comparisons of brightness and histological assessment of expression (data used for analysis in Figure 5, and FOV and virus expression shown in Figure S4d).

Specific points

How does the number of labeled neurons (perhaps measured by the ratio of anti-GFP to NeuN labeling) depend on the GCaMP type (Fig. 2b)?

A quantification of overlap between NeuN and RO injected php.eb virus with a Synapsin promoter has been performed in earlier work (e.g. Chan et al., 2017).

The focus of our work has been in vivo performance. We have therefore included a quantification of the number of detected ROIs (active cells) for the different GECIs, which is now shown in Figure 5.

The number of labeled cells looks quite different in Ribo and soma images (Fig. 3). Is this because the firing is different or there are different number of neurons labeled. Do the really bright cells still have normal firing?

The number of cells detected between Ribo and EE-RR is indeed different. We believe this is mainly an effect of the almost complete lack of neuropil signal with ribosome-

tethering, which masks cell signals for the other GECIs. A similar result has been shown previously with the first version of Ribo-GCaMP(6m) (Chen et al. 2020).

Please also clarify the promoter used in each experiment.

An hSyn (Synapsin) promoter was used in all GECI experiments, this is now clarified in the text.

For soma-targeted GCaMP (again especially for the new constructs), please show higher magnification images of the novel soma targeting versions to show where the expression is. Would likely need a higher NA objective than 0.8NA used here.

This has been added to a new version of Figure 4. We have tracked specific populations from 2-10 weeks using a 20X 1.0NA objective.

Also need quantification of change in fluorescence from the soma versus the surround, rather than just the base-line brightness.

As noted for the first major point, this has been added to Figure 5.

In widefield imaging, how were potential variations in activity due to different amounts of running or other behaviors accounted for across the mice?

We have not accounted for running or other such behaviors in widefield imaging. This method was mainly used to screen for potential seizures.

Fig. 1C and D: Please state which GECI was used in the example

This has been corrected.

Intro: "Overexpression in a subset of cells can lead to intracellular aggregation and eventually cell death." Did you mean that cells are protected when there is overexpression everywhere or that overexpression in any number of cells is problematic?

Overexpression in cells in itself is problematic, as it will often interfere with signals from other ROIs when performing automated analysis. In addition, overexpression of GECIs likely affects cellular function. This has been clarified in the text.

It not really neuropil “contamination”, as the neuropil signal itself can be useful. Perhaps there is better wording for this as often the signal does reflect true neuronal activity.

We acknowledge that neuropil signal likely reflects neuronal activity, and have softened the language where appropriate. Yet, when only somatic signal is considered, which is often the case for 2P GECI imaging, neuropil signal of unknown origin (likely a mix of different neurons) can interfere with somatic measurement; ie. contaminating the somatic signal.

Please define MOI in text better and move the speculation about the mechanisms to Discussion. Otherwise, it would be good to measure the viral vector genetic material in the cells.

MOI clarified in text and moved to discussion.

Measurements of viral content would be helpful to include, but represents a considerable amount of additional work. As a lower concentration of viral particles is delivered to the tissue from a systemic administration via the blood versus a local injection of high titer aav directly into the tissue, a lower MOI is to be expected - resulting in both reduced expression and a reduction in the total number of expressing cells. Additionally, it is not the goal of this paper to characterize the PHP.eB serotype, which is already well characterized.

Do you see GCaMP accumulation in the surrounding tissues such as the in the dura and if so, was it different with the various delivery options?

We have not observed any GCaMP accumulation in the dura or other surrounding tissues. As far as we are aware PHP.eB AAV is not known to infect cells in the dura, and in general primarily infects neurons.

“In an attempt to improve the brightness, we tested both double and triple injection volumes of RO administered GCaMP6f, but the resulting expression was still too dim to image at reasonable laser power (data not shown).” Showing even supplementary data on this would be useful for the community so other people don’t have to try it.

We have included histology from a triple injection of jGCaMP8f in Figure 2 (2-photon data is similar to the single 6f injection, i-e little to no signals to image at all).

In Results: “the expression remained stable across weeks (Fig. 1C), with no indication of intracellular aggregation.” It looks like it is increasing in brightness at 6 weeks.

We have added more time-points to Figure 1. Additionally, we have included staining and quantification of astrocytic activation (Fig S5), and added high-resolution images from 2, 4, 6, 8 and 10 weeks after injection (Fig 4).

Fig. 2b caption – please also indicate that this is GFP antibody labeling in caption.

This has been corrected.

Fig. 3b – are the traces shown from particular neurons indicated on the left? Hard to tell from the color code in left panel.

The traces have been removed to make way for new data, but traces from EE-RR and RiboL1 versions of jGCaMP8s are included in Fig S5, with the cells indicated in the image.

In Results: “local injections of two or more viruses often leads to competition and low co-expression” – Please provide substantiation for this statement or eliminate. There’s lots of anecdotal evidence that this works quite well.

It is difficult to find published data on this issue, perhaps not surprisingly as negative results rarely get published. But in our hands and in the experience of many other labs we have discussed the phenomena with, it can indeed be challenging to express two viruses in the same cell. However, we agree that this statement was somewhat strong, and have removed it in place of another argument for RO injections in combination with other cell-type targeted constructs.

“flexed hM4D DREADD receptor” should this be “floxed”?

From our understanding, floxed usually refers to flanking a sequence with single LoxP sites facing 5'-3', which will result in excision of the flanked sequence in reaction with Cre. Here, we are referring to a FLEX plasmid, ie. a sequence surrounded by multiple LoxP sites oriented such that the sequence is inverted, not removed in reaction with cre. This is also known as DIO. We erroneously referred to this as floxed earlier in the paper, a typo which has now been corrected. FLEX-hM4D receptor, or FLEXed hM4D receptor are probably more accurate than “flexed”.

In discussion “An additional caveat concerning the PHP.eB and AAV9 is that we find a clear bias in expression for cortical layer 5, striatum, CA2 and subiculum regions.” Please show a quantification of this. The effect is not so clear that I can tell from the images.

The data we refer to was included in the Supplementary Data, that we regrettably had not uploaded. It is now included as Figures S2 and S3. In addition, these effects are well documented in other papers (e.g. Chan et al., 2017; Mathiesen et al., 2020).

Metacam/meloxicam: please use consistent wording

This has been corrected.

Were imaging sessions awake or anesthetized? Please provide details for either (e.g. training for awake imaging)

All experiments were performed in awake mice on a running wheel. This has been clarified in the text, and a paragraph on habituation to the apparatus has been added to the Methods section.

REVIEWER COMMENTS

Reviewer #1 (Remarks to the Author):

The authors have addressed all of my concerns, which largely concerned lack of quantification and replication.

Reviewer #2 (Remarks to the Author):

I am satisfied with the response of the authors to my specific comments. The additional quantifications and increases in animal numbers has improved the significance of this work. I recommend this paper for publication.

Reviewer #3 (Remarks to the Author):

Some concerns have been met with new data. For example, showing the same cells over 10 weeks does demonstrate limited toxicity. However, a major concern remains that most the data are about expression, and it very difficult to gauge what is the functionality of each of the labeling strategies. Expression data and baseline fluorescence are not necessarily related to functional metrics. While there has been addition of the data in Fig. 5 such as the soma/neuropil correlation, it is still hard to tell whether there would be differences in how the different labeling strategies respond to similar stimulation or show variations in the reported spontaneous firing rate. For example, while independent measures of neural activity like ephys would be best, a demonstration of the same stimulus applied to different labeling strategies with quantifications like number of calcium events, duration, and magnitude could be enough for a user to decide whether these indicator strategies are sufficient for their experiments.

Lack of this still limits the usefulness of this paper.

RESPONSE TO REVIEWER COMMENTS

We thank the reviewers for their helpful comments to our manuscript. We have addressed the final comments from Reviewer #3 by adding new analysis regarding stimulus-driven and spontaneous activity.

Below are the reviewers' comments with our replies in blue text.

Reviewer #1 (Remarks to the Author):

The authors have addressed all of my concerns, which largely concerned lack of quantification and replication.

Reviewer #2 (Remarks to the Author):

I am satisfied with the response of the authors to my specific comments. The additional quantifications and increases in animal numbers has improved the significance of this work. I recommend this paper for publication.

Reviewer #3 (Remarks to the Author):

Some concerns have been met with new data. For example, showing the same cells over 10 weeks does demonstrate limited toxicity. However, a major concern remains that most the data are about expression, and it very difficult to gauge what is the functionality of each of the labeling strategies. Expression data and baseline fluorescence are not necessarily related to functional metrics. While there has been addition of the data in Fig. 5 such as the soma/neuropil correlation, it is still hard to tell whether there would be differences in how the different labeling strategies respond to similar stimulation or show variations in the reported spontaneous firing rate. For example, while independent measures of neural activity like ephys would be best, a demonstration of the same stimulus applied to different labeling strategies with quantifications like number of calcium events, duration, and magnitude could be enough for a user to decide whether these indicator strategies are sufficient for their experiments. Lack of this still limits the usefulness of this paper.

As a general note to the different quantifications that the reviewer suggests, there is no reason to think these should be affected by the delivery route of the virus (see for example Madisen et al., 2016; Hamodi et al., 2020), other than the effects already shown in Figure 5. The different delivery routes will only influence the number of virus particles taken up by each cell (and thus characteristics such as baseline brightness, shown in Fig 5b), while the identity of the GECI will determine the response kinetics such as sensitivity, duration and magnitude. These have been mapped out in detail by others (e.g. Dana et al., 2019; Zhang et al., 2020).

We have therefore added analyses to Fig S5 that compares the spontaneous and stimulus-evoked responses between the two soma-targeting strategies with the unaltered jRCaMP1a. Our new analyses show that neurons transfected with the RiboL1 construct show somewhat higher spontaneous and stimulus-evoked activity rates and quite high variability. We believe that this results from the fact that a much larger fraction of the population is labeled using this construct, as shown in Fig 5, and thus representing a more complete picture of what the population is doing. This is supported by the fact that using the same construct with a systemic injection, which yields a smaller population of detectable cells per animal, the same parameters are not significantly different from the other groups.

However, it should also be noted that our experiments were not specifically designed to test detailed parameters of each GECI. For example, we have injected different viruses in separate parts of the very same cranial window with little consideration for retinotopic location or monocular/binocular separation. Moreover, the activity in the cortex is affected by many different behavioral parameters such as locomotion and attention, and as a result of screening many different GECIs in parallel (up to 15 mice imaged in the same day), the recorded data per animal in our study is somewhat limited (5-10 minutes of recording per time point).

REVIEWERS' COMMENTS

Reviewer #3 (Remarks to the Author):

The revised manuscript has addressed my concerns.